# Switchable unidirectional emissions from hydrogel gratings with integrated carbon quantum dots

Chenjie Dai[1], Shuai Wan[1], Zhe Li[1], Yangyang Shi[1], Shuang Zhang [2] ✉ & Zhongyang Li [1,3,4,5] ✉

Directional emission of photoluminescence despite its incoherence is an attractive technique for light-emitting fields and nanophotonics. Optical metasurfaces provide a promising route for wavefront engineering at the subwavelength scale, enabling the feasibility of unidirectional emission. However, current directional emission strategies are mostly based on static metasurfaces, and it remains a challenge to achieve unidirectional emissions tuning with high performance. Here, we demonstrate quantum dots-hydrogel integrated gratings for actively switchable unidirectional emission with simultaneously a narrow divergence angle less than 1.5° and a large diffraction angle greater than 45°. We further demonstrate that the grating efficiency alteration leads to a more than 7-fold tuning of emission intensity at diffraction order due to the variation of hydrogel morphology subject to change in ambient humidity. Our proposed switchable emission strategy can promote technologies of active light-emitting devices for radiation control and optical imaging.

Emission steering is an important technique for the practical applications of light-emitting materials, including quantum dots (QDs)[1], perovskite[2], rare-earth doped materials[3], and light-emitting diodes (LEDs)[4]. Different from the coherent source, the typical photoluminescence (PL) exhibits omnidirectional emission with random phases and negligible spatial coherence, leading to inevitable challenges in terms of its wavefront engineering and radiation control. The emerging field of nanophotonics, and in particular metasurfaces, creates freedom for controlling the propagation of light[5–9], and offers a promising route to enhance and control light emission[10–15].

Light-emitting metasurfaces are typically composed of nanoscale emitters and two-dimensional meta-atoms that can directly manipulate the far-field diffraction of the localized emission[16,17]. To flexibly customize the functionalities of emission, light-emitting metasurface has been investigated to achieve emission enhancement[18,19], chiral emission[20–22], and unidirectional emission[23–25] via the principles of localized resonances[26,27], grating effect[28], bound states in the continuum[20,29], etc. For example, though embedding the quantum wells above the phase-gradient metasurface, the PL could be steered to the desired diffraction order with a divergence angle of ≈7° [23]. On the one hand, the integration of light-emitting materials and metasurface enables attractive functional emission, however, most proposed strategies are static directional control without any tunability or spectral tuning, thus limiting their practical application scenarios in optical displays and optical communications. On the other hand, although various reconfigurable gratings and metasurfaces have been employed for dynamic beam steering with coherent external sources[30–33], their integration with light-emitting materials has still not been fully explored.

To explore the dynamic tuning ability of directional emission using reconfigurable nanophotonic devices, the spatial light modulator (SLM) has been recently exploited to modulate the amplitude

[1]Electronic Information School, Wuhan University, Wuhan 430072, China. [2]Department of Physics, The University of Hong Kong, Hong Kong 999077, China. [3]Wuhan Institute of Quantum Technology, Wuhan 430206, China. [4]School of Microelectronics, Wuhan University, Wuhan 430072, China. [5]Suzhou Institute of Wuhan University, Suzhou 215123, China. ✉e-mail: shuzhang@hku.hk; zhongyangli@whu.edu.cn

distribution of pump light in real-time and create continuous phase gradients onto the semiconductor metasurface, proving the feasibility for dynamic emission steering[34]. While such a spatiotemporal control system exhibits fast and dynamic emission control, the maximum diffraction angle is restricted to 35°, and the divergence angle is greater than 3° due to the limitations of complex optical set-up and the SLM pixel size. Thus far, there has been no efficient and feasible strategy for light-emitting nanophotonic devices to achieve switchable unidirectional emission with a wide diffraction angle and narrow divergence angle.

Here, we experimentally demonstrate switchable and unidirectional emission with a large diffraction angle (>45°) and an ultranarrow divergence angle (<1.5°) from visible-frequency quantum dots-hydrogel integrated gratings (Q-HIG). By employing polymer shrinkage under the e-beam exposure, the QDs-embedded polyvinyl alcohol (PVA) hydrogel film can be directly fabricated for asymmetric grating profile with single-step processing, imparting the grating momentum to the PL emission and allowing the PL in the guided mode to be unidirectionally coupled out into free space. Thanks to the humidity-responsive swelling capability of hydrogel materials[35–39], the grating efficiency of hydrogel grating could be notably tuned with ambient humidity change due to the morphology alteration, enabling efficient control of emission, beyond typical grating schemes for static emission extraction[2,40]. Furthermore, the excitation position creates extra degree of freedom to manipulate the emission direction of the PL. In addition, we demonstrate a quantum dots-hydrogel integrated nanocavity for tunable directional emission to prove the potential of hydrogel nanophotonic devices for versatile emission tuning. Such a compact and switchable unidirectional emission platform featured with simplified fabrication and easily accessible tuning strategy could facilitate technologies of active emission tuning and suggests the potential applications in several fields, including biomedical imaging, holographic displays, and optical communications.

## Results
### Performance of quantum dots-hydrogel integrated gratings

Figure 1 represents the schematic view of the Q-HIG construction and its active unidirectional emission control strategy. The commercial carbon quantum dots (CQDs) are randomly embedded in the hydrogel grating on a reflected Ag substrate (Fig. 1a). Under the 532-nm pump excitation, a small part of PL from QDs directly emits into free space, and the majority is confined to the guided modes with in-plane

wavevector $\mathbf{k}_{//}$ and propagates in the hydrogel film due to the critical angle restriction[4,23]. The in-plane momentum $\mathbf{k}_{//}$ is related to the propagation angle $\theta_i$ in the hydrogel and could be represented as

$$\left|\mathbf{k}_{//}\right| = n_{PVA}\,\sin\theta_i k_0 \approx k_0\, n_{PVA}\sqrt{1 - \left(q\lambda/(2n_{PVA}h)\right)^2} \qquad (1)$$

where $n_{PVA}$ is the refractive index of PVA, $q$ is the mode number, $h$ is the thickness of PVA and $k_0 = 2\pi/\lambda$ is the free-space momentum of the light wavelength $\lambda$. When the in-plane emission passes through the grating, the hydrogel grating imparts a momentum $\mathbf{k}_G = 2\,m\pi/P$ onto the guided modes[41,42], where $m$ is the diffraction order, enabling the direction control of PL. To steer the guided light into free space, momentum accumulation is required to satisfy the condition $|\mathbf{k}_{//} + \mathbf{k}_G\,|/k_0 < 1$.

If the pump beam is directly incident onto the grating region, due to the positive/negative momentum combination from the grating, the guided PL would be emitted into symmetric diffraction orders (Fig. 1a). To break the symmetric radiation, one can change the excitation position to one side of the grating region (Fig. 1b), which allows the unidirectional emission of PL into a single diffraction order. Moreover, the radiation energy of unidirectional emission could be dramatically tuned by increasing the relative humidity (RH) due to the grating efficiency decline from hydrogel morphology alteration.

### Hydrogel grating fabrication and characterization

We first characterize the PL performance of the spin-coated CQDs-integrated hydrogel thin film. The measured PL spectra of thin film range from 550 nm to 680 nm with an emission peak of 575 nm under the 532 nm pump (Fig. 2a). The thin film was spin-coated to ≈625 nm thickness at 3000 rpm for next-step fabrication (Fig. 2c). The initial thickness of spin-coated films depends on the rotation speed and the PVA concentration. To construct periodic grating on the QDs-embedded hydrogel thin film, the grayscale e-beam lithography (G-EBL) is employed to directly sculpt the desired profile due to the does-induced polymer shrinkage (Fig. 2b), as demonstrated in our previous work[43]. In the experiment, we fabricated Q-HIG with different periods and measured their geometric profile using an atomic force microscope (Fig. 2d). The angle-resolved reflections of the gratings under the normal incident confirm the capability of the grating to steer light into diffraction orders (Fig. 2e). Because the adopted exposure dose is gradient and asymmetric along the $x$-direction, the fabricated gratings inherit those properties and enable asymmetric optical performance in the far-field diffraction. To further characterize the

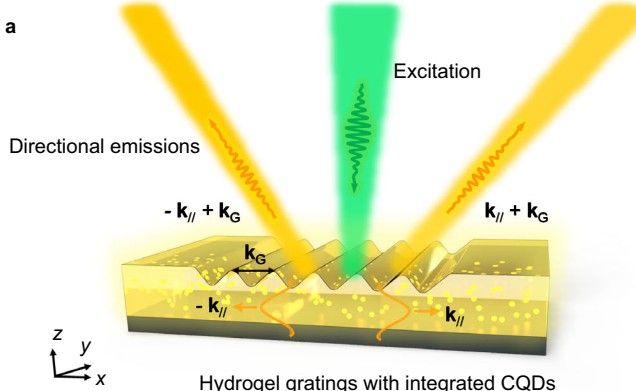

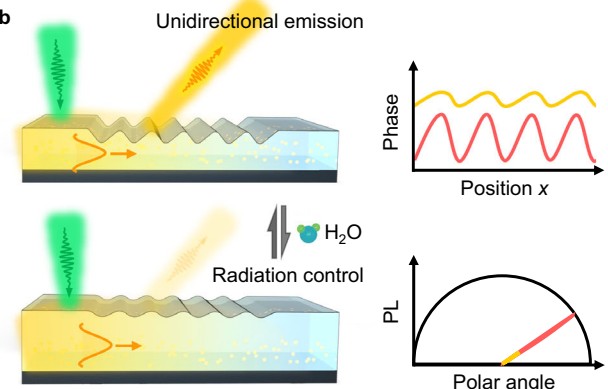

**Fig. 1 | Design of QDs-hydrogel integrated grating for directional PL emission and radiation control. a** Schematic of the designed architecture for steering the guided mode emission into free space. The in-plane momentum $\mathbf{k}_{//}$ of QDs emission could be redirected and coupled to leaky waves due to the grating momentum $\mathbf{k}_G$ accumulation from hydrogel grating. The excitation wavelength is 532 nm (green) and the PL peak wavelength is 575 nm (orange). **b** Operation principle for

unidirectional emission and radiation control. The spatial location of the excitation manipulates the direction of in-plane momentum for passing through the grating. The grating efficiency could be tuned by changing the relative humidity around the sample due to the hydrogel morphology alteration, enabling the radiation energy switch. The inset represents the phase change of hydrogel grating and the PL energy variation under middle RH (red) and high RH (golden) conditions.

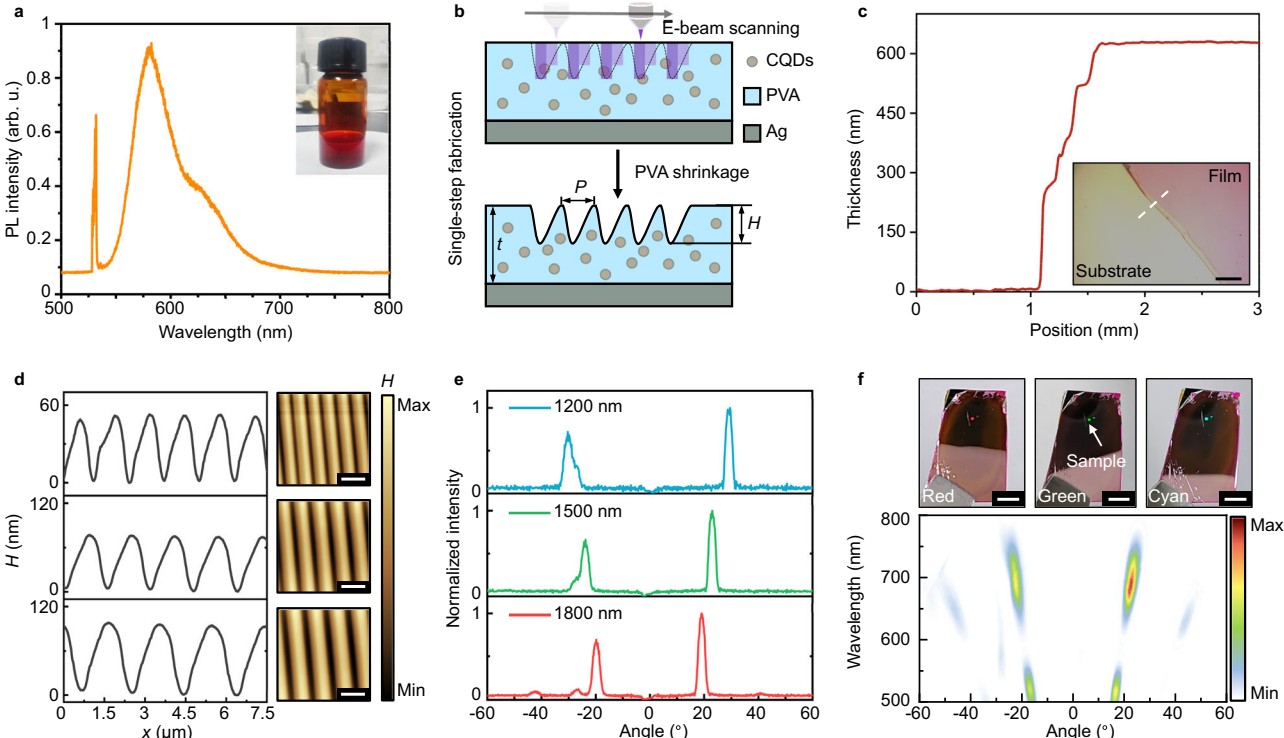

**Fig. 2 | Fabrication of Q-HIG and its optical properties. a** Spectral measurement of PL from CQDs-integrated hydrogel film. Inset: The mixture is created by dissolving solid PVA powder into deionized water and then mixing it with CQDs aqueous solutions. **b** Schematic of the single-step fabrication of Q-HIG using grayscale e-beam lithography. **c** Measured thickness of PVA-CQDs thin film spin-coated at 3000 rpm. Scale bar, 3 mm. **d** Atomic force micrographs of the fabricated Q-HIG with periods of 1200 (blue), 1500 (green), and 1800 nm (red), respectively. Scale bar, 2 μm. **e** The measured angle-resolved reflection of gratings under the corresponding periods at the wavelength of 575 nm. **f** Measured broadband angle-resolved reflection of the grating with an 1800-nm period. The top photographs captured under different shooting angles show the angular dispersion of the grating. Scale bar, 2.5 mm.

operating bandwidth of gratings, the angle-resolved reflection is measured under white light illumination, which confirms the broadband properties of the Q-HIG (Fig. 2f). In addition, the asymmetric far-field diffraction could be further enhanced by increasing the asymmetry of the grating structure. However, there are some challenges in fabricating perfect blazed hydrogel gratings of this size using current PVA direct printing technology. We speculate that perfect blazed hydrogel gratings could be fabricated by using the nanoprinting technique or improving the material properties.

### PL experiments of hydrogel grating with CQDs

To demonstrate directional emission of the Q-HIG, we employ an unpolarized green (532 nm) laser pointer as the pump source to excite the CQDs for PL emission. Figure 3a shows the optical microscopy images of the fabricated grating under the white source illumination and pump excitation. To study the directional PL from grating, the far-field angular emission distributions of the QDs-embedded film and grating are measured in the back focal plane under the same excitation conditions (Fig. 3b, c). Compared to the film's omnidirectional emission, the 1800-nm period grating deflects the guided-mode PL to the desired order with a narrow divergence angle and a significant intensity boost, as plotted in Fig. 3d. Based on the grating momentum $\mathbf{k}_G$ and in-plane momentum $\mathbf{k}_{//}$, the PL emission is identified to be the second diffraction order of the grating ($m = \pm2$) while the first diffraction order ($m = \pm1$) is not satisfying the outcoupling condition, and the PVA refractive index is numerically retrieved to ≈1.52, slightly higher than the pure PVA film's refractive index of ≈1.51 due to QDs integration (see Supplementary Figs. 3, 4). Furthermore, the polarization of the emitted PL is primarily along the $y$-axis (S-pol.), which results from the interplay between the in-plane emission mode and the grating construction

(Fig. 3e). By comparing the PL intensity between the grating and a film at the corresponding order angle under the same illumination condition, we find an over-tenfold directional PL enhancement of Q-HIG (Fig. 3f). In addition, the divergence angle $\Delta\theta$ of the enhanced order is measured to be less than 2° in the broadband visible regime (see more details in Supplementary Fig. 5), which is superior to most current directional emission strategies.

### Humidity-responsive beam steering and PL emission

We further demonstrate switchable emission ability of the hydrogel platform via humidity-driven grating efficiency alteration on the beam steering and PL emission. In our experiment, the ambient humidity is controlled by blowing dry or wet nitrogen gas onto the sample using a gaseous injector, as shown in Fig. 4a. By increasing the ambient humidity from middle RH (40–60%) to high RH (70–90%), the efficiency of Q-HIG for steering the normal incident light into the relevant diffraction order is eliminated due to the hydrogel morphology change (Fig. 4b). This agrees with our numerical simulation of the angle-resolved reflection change under different humidity conditions by considering the hydrogel inflation based on the previous measurement[43,44], as shown in Fig. 4c (Supplementary Fig. 6). Under a high RH, both the simulation and experimental measurement show that the diffraction orders disappear due to the elimination of the phase gradient. This dramatic change controlled by humidity ensures the feasibility of active emission control. To further demonstrate the humidity-responsive PL tuning, the far-field angular distributions of PL are measured at middle RH (Fig. 4d) and high RH (Fig. 4e) under the unpolarized 532-nm pump. When the sample is exposed to high RH > 70%, there is nearly no conversion from guided PL emission into the free space due to the very weak grating contrast. Figure 4f plots the PL angular intensity profile at 632 nm to compare and verify the

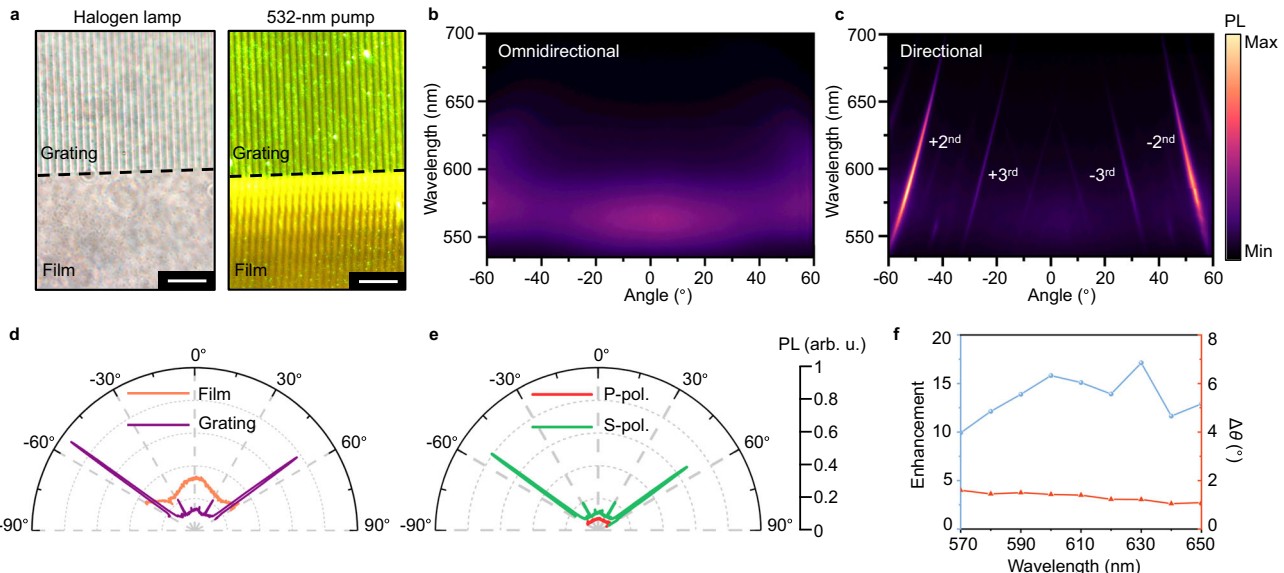

**Fig. 3 | Directional PL emission from Q-HIG. a** Top-view of the fabricated hydrogel grating under the illumination of the halogen lamp and 532-nm pump. Scale bar, 10 µm. **b, c** Measured angle-resolved dispersion diagrams of PL emission from PVA-CQDs (**b**) thin film and (**c**) grating, respectively. **d** Corresponding far-field radiation polar plots of PL for films (orange) and gratings (purple) at a wavelength of 575 nm. **e** Measured the polarization component of PL for s-polarization (S-pol.) and p-polarization (P-pol.) using an analyzer. **f** Calculated enhancement and divergence angle $\Delta\theta$ of the directional PL emission. The value of enhancement is calculated by $I_{grating}/I_{film}$, where $I_{grating}$ is the PL intensity of the grating at the diffractive order, and $I_{film}$ is the PL intensity of the film at the same angle. The divergence angle $\Delta\theta$ is defined by the full width at the half-maximum of the emission angle.

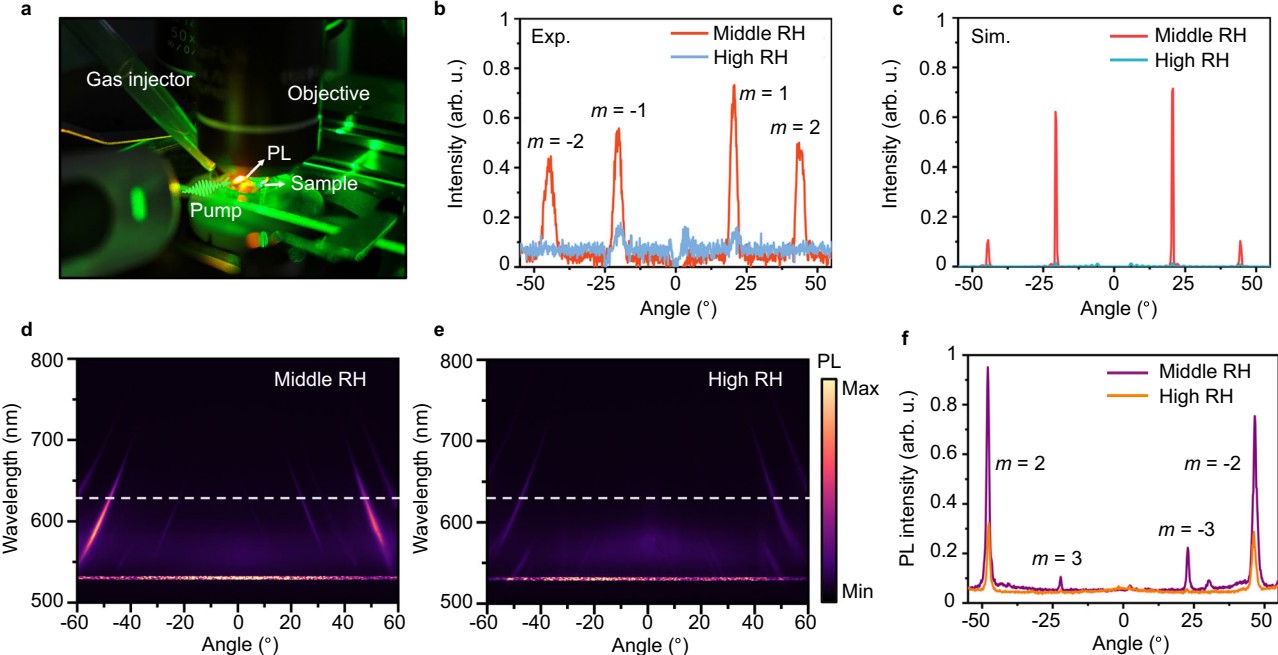

**Fig. 4 | Humidity control of PL radiation. a** Experimental setup for the PL measurement with humidity applied. **b** Measured angle-resolved reflection intensity alteration under the middle RH (red) and high RH (blue) conditions at the normal incidence. **c** Simulated the angular intensity change from the morphological transformation of hydrogel grating. **d, e** Experimentally measured far-field angular dispersion of PL emission from Q-HIG under the middle and high RH, respectively. **f** Line plots of the corresponding PL intensity in (**d, e**), as marked by dashed lines.

switchable emission functionalities. Based on the theoretical calculation, the emission peak around −50° is 2nd order, and the peak at −25° around 3rd order (Supplementary Fig. 4).

### Switchable unidirectional emission

We further show that the emission can be directed into a single direction by tailoring the momentum direction of guided PL, as schematically depicted in Fig. 5a. By adjusting the pump position to the left side of grating, the guided mode PL mostly propagates through the grating architecture with a positive momentum $\mathbf{k}_{//}$ and is imparted with a second-order grating momentum $\mathbf{k}_G$ ($m = -2$), resulting in the unidirectional emission of PL. As anticipated, the PL is steered to the desired order in our measurement (Fig. 5b). Moreover, the intensity of unidirectional PL also exhibits intense reduction under the high

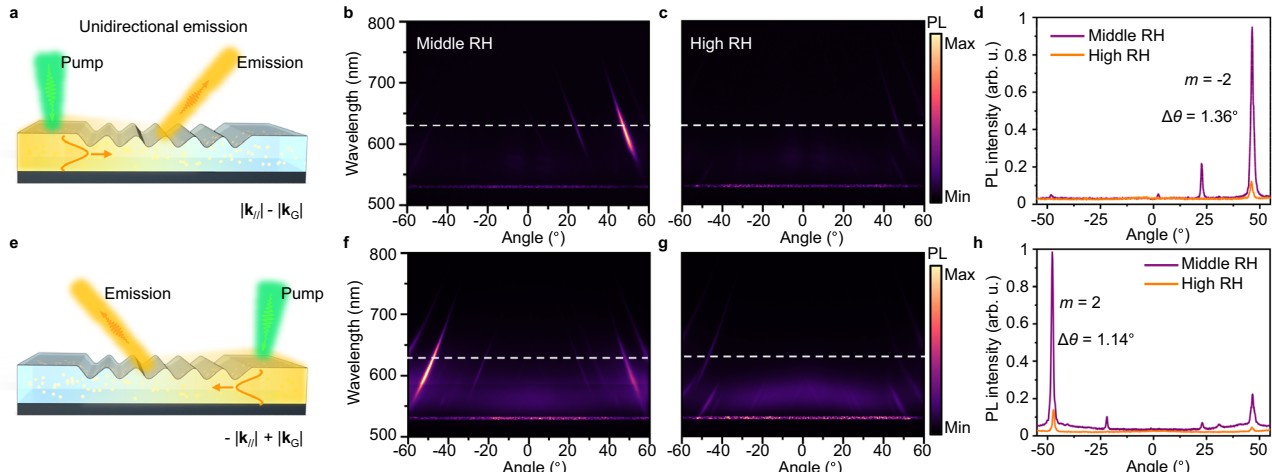

**Fig. 5 | Unidirectional emission and radiation control. a** Schematic of the unidirectional PL from Q-HIG. The hydrogel grating steers the PL toward the diffraction order with the far-field emission momentum $|\mathbf{k}_{//}| + |\mathbf{k}_G|$ ($m = -2$) under the left excitation. **b, c** Measured angle-resolved PL intensity under the middle and high RH conditions. The intensities of PL measurements are normalized to the same scale. **d** Line plots for corresponding measured PL intensities in (**b, c**) at 632-nm wavelength (dashed lines). **e** Schematic of the PL emission with the emission momentum $-|\mathbf{k}_{//}| + |\mathbf{k}_G|$ ($m = 2$) under the right excitation. **f, g** Measured angle-resolved PL intensity (**f**) enhancement and (**g**) disappearance at different RH conditions. **h** Line plots for corresponding PL intensities alteration in (**f, g**) with a narrow divergence angle of 1.14°, as marked by dashed lines.

RH due to the efficiency alteration from hydrogel inflation (Fig. 5c). Figure 5d compares PL intensity change under the different RH conditions at the wavelength of 632 nm, revealing the switchable unidirectional PL with approximately 7-fold radiation control and high directionality ($\Delta\theta = 1.36°$). By changing the pump position to the right side of grating (Fig. 5e), the measured far-field distributions show the expected direction reversal under the middle RH, and again, the radiation intensity decreases under high RH (Fig. 5f, g). The corresponding PL intensity profiles plotted to verify the switchable unidirectional PL emission, as shown in Fig. 5h. In addition, we demonstrate the repeatability of the Q-HIG performance for switchable emission during cycle experiments (Supplementary Fig. 10). When the ambient environment around the grating cyclically alternates between middle and high RH, the PL intensity at the diffraction order could be reversibly tuned as well as other PVA-based devices[36,43]. The integration of hydrogel and QDs may provide a guideline for future light-emitting nanodevice design, and this methodology may be adapted to realize dynamic PL functionalities, such as tunable focusing and dynamic wavefront engineering.

### Hydrogel nanocavity for tunable directional emission

To prove the potential of hydrogel-based nanophotonic devices for versatile emission tuning, we further demonstrate a quantum dots-integrated hydrogel nanocavity (Q-IHN) for tunable directional emission (Fig. 6a). Because the emission direction cannot be directly tuned by humidity based on hydrogel grating schemes, we employ the angle-dependent resonance from Fabry-Perot type nanocavity to tailor the direction of PL emission. With the RH increase, the angle-dependent resonance actively shifts and leads to a continuous emission angle variation from 0° to 40° due to the cavity-induced angular enhancement of the Purcell effect and excitation rate[39,45]. Figure 6b shows the measured angular-resolved PL emission under different RH conditions. The emission angle is distinctly tuned by increasing RH due to the angle-dependent absorption shift from hydrogel layer inflation (see more details in Supplementary Figs. 11–14). When the RH increases from ≈60% to 85%, the emission angle actively transforms from 0° to 40° at the emission wavelength of 575 nm (Fig. 6c), revealing the tunability of the QDs-integrated hydrogel platform for continuous directional emission control.

## Discussion

In summary, we have experimentally demonstrated a humidity-driven integrated platform for switchable unidirectional PL emission by incorporating light-emitting materials with hydrogel-based grating. By exploiting the PVA shrinkage under e-beam exposure, we present a single-step fabrication approach to construct the proposed grating with different periods. The hydrogel grating enables directional emission of the guided PL from embedded QDs and provides tunability of grating efficiency modulation, allowing the dynamic control of far-field emission. In addition, we prove the potential of hydrogel-based platforms for tunable direction emission using a Fabry-Perot type nanocavity. Our proposed strategy for actively emission tuning based on hydrogel materials could serve as a promising guideline of active light-emitting metasurfaces for potential applications, including optical display, biological imaging, and incoherent light source.

## Methods

### Numerical simulations

The finite-difference time-domain (FDTD) method was utilized to calculate the far-field angular distribution of the hydrogel gratings at the normal incident. 3D simulations were performed for the grating with a unit cell area of $100 \times 1800$ nm² at the $x$-$y$ plane using periodic boundary conditions, with perfectly matched layers along the propagation of electromagnetic waves ($z$-axis). The refractive index of hydrogel is set as 1.51, and the complex refractive index of Ag is utilized from the data of Palik (0–2 μm)[46].

### Sample fabrications

For the PVA-CQDs aqueous solution, the commercial hydrolyzed PVA (Thermo Fisher Scientific) with the average molecular weight $M_w$ of 10–26 kg mol⁻¹ was firstly dissolved in deionized water for the mass percentage concentration of 10.7 wt%. Subsequently, the CQDs aqueous solution purchased from Mesolight Inc (Suzhou, China) with a concentration of 10 mg mL⁻¹ was mixed with the prepared PVA aqueous solution. The mixing ratio of PVA and CQDs aqueous solutions is 9:1. Ultimately, the PVA-CQDs solution was spin-coated onto the Ag substrate at the speed of 3000 rpm for 60 s to give a ≈625 nm thick film. The QDs-embedded hydrogel film was directly exposed by G-EBL (Raith eLINE Plus, electron acceleration voltage of 10 kV) for hydrogel gratings fabrication.

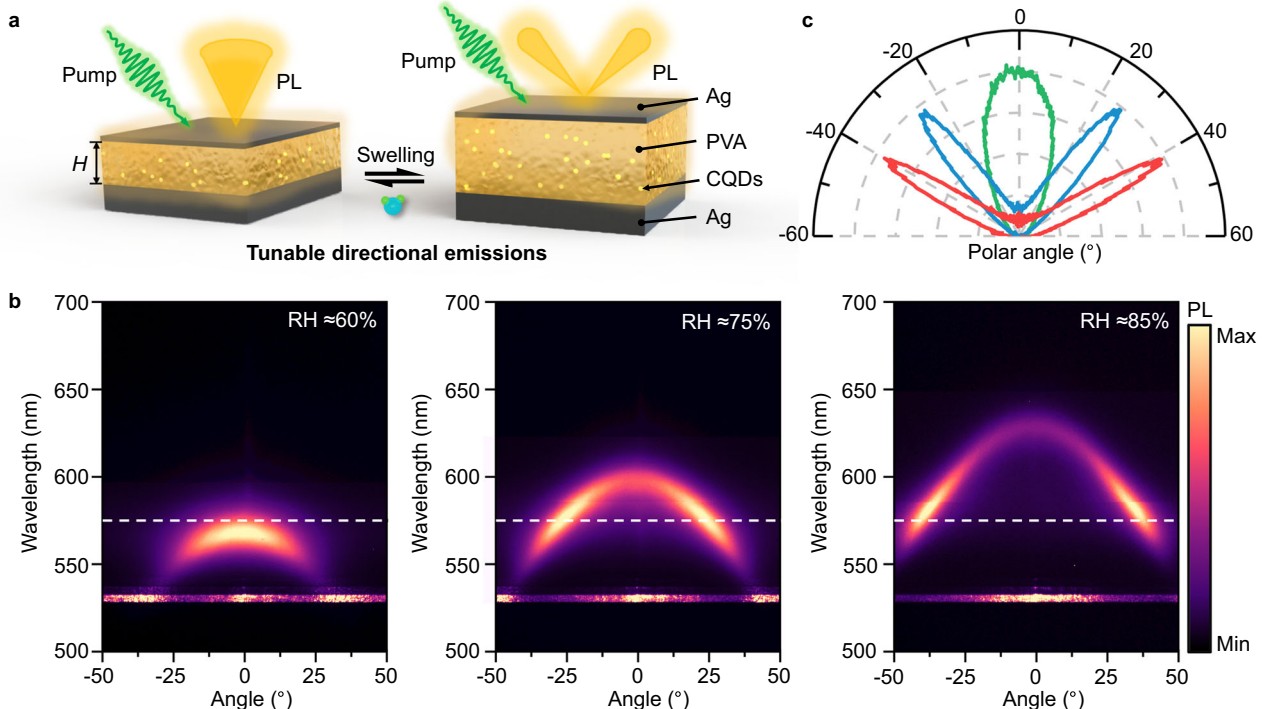

**Fig. 6 | Tunable directional emissions. a** Graphical illustration of quantum dots-integrated hydrogel nanocavity for tunable directional emission. **b** Experimentally measured far-field angular distribution of PL emission from Q-IHN under different RH. **c** Polar plots for corresponding angle-resolved PL intensity in (**b**) at the wavelength of 575 nm under the RH of ≈60% (green), 75% (blue), and 85% (red), as marked by dashed lines.

## Optical measurement

The Q-HIG was obliquely pumped with unpolarized green (532 nm) laser pointer for PL emission. The PL was collected through a 100× objective, and its angular intensity distribuend at Fourier back focal plane is measured by an angle-resolved microscopic spectrometer based on the Fourier transform (built by Ideaoptics Inc.). For the humidity test, the PL measurement at middle RH condition (RH 40–60%) is directly performed under indoor humidity. Regarding the high RH condition, a flow of dry nitrogen is bubbled through deionized water in a sealed bottle to generate wet gas with an RH of 70–90%, and the wet gas is applied to the sample surface for high RH by using a gaseous injector.

## Reporting summary

Further information on research design is available in the Nature Portfolio Reporting Summary linked to this article.

## Data availability

The data that support the findings of this study are available from the corresponding authors upon request.

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

## Acknowledgements

Zho. L. gratefully acknowledges the National Key Research and Development Program of China (No. 2022YFB3808600), National Natural Science Foundation of China (No.12204360), the Fundamental Research Funds for the Central Universities (2042023kf0227), Hubei Province Funds for Distinguished Young Scientists (2021CFA043). S.Z. gratefully acknowledges the New Cornerstone Science Foundation, the Research Grants Council of Hong Kong (AoE/P-502/20 and 17315522). The authors thank the Core Facility of Wuhan University for the measurements of AFM. The work was supported by the Center for NanoScience and Nanotechnology at Wuhan University.

## Author contributions

Zho. L. and C.D. conceived the idea. C.D. and S.W. implemented the simulations and fabricated the samples. C.D., Zhe. L., and Y.S. performed the optical measurement. Zho. L. and S.Z. supervised the project. C.D., Zho. L. and S.Z. drafted the manuscript and all authors discussed the results and revised on the manuscript.

## Competing interests

The authors declare no competing interests.
