## [Peer Review File · Nature Communications]

Switchable unidirectional emissions from hydrogel gratings with integrated carbon quantum dotsReviewers' comments:

Reviewer #1 (Remarks to the Author):

Please see attached file for comments.

Review report on paper titled ‘Tunable unidirectional emission from quantum dots–hydrogel integrated metagratings’ by Chenjie Dai, Shuai Wan, Zhe Li, Yangyang Shi, Shuang Zhang, and Zhongyang Li

The manuscript reports on the optical properties of a polymer grating incorporating light-emitting quantum dots. The key claimed result is the demonstration of directional emission and the ability to change the emission pattern by changing the relative humidity. Light-emitting devices with bright directional emission in a controllable direction are key components in a wide range of applications. The topic of the present study is thus of clear importance. The use of metasurfaces, two-dimensional arrays of artificial ‘meta-atoms’, is a promising and intensively studied approach to control light emission. The topic of the study could thus have a large audience.

The study is in general well performed and the technical claims are supported by the data. The paper is mainly well written and the illustrations are clear. However, I do not agree with the claimed novelty and importance of the results. First of all, the authors call their structure a ‘meta-grating’ trying to place their structure in the area of metamaterials and metasurfaces. Their device is clearly a (blazed) grating and should be called such. The directional emission is based on out coupling from the polymer layer using the grating, which is a standard approach. The performance of the grating is good but there is no novel concept involved. The main novelty appears to be the use of elevated humidity to quench the grating and thus the emission. Overall, the concept is not a major and general advance of the field but rather of technical and specialized nature. Finally, there is a plethora of studies on reconfigurable gratings and metasurfaces, for example, *Nature Commun.* **9**, 1396 (2018); *Small*, 2301871 (2023); *Nano Lett.* **21**, 1238 (2021); *Adv. Optical Mater.* **7**, 1801786 (2019); and *Science* **364**, eaat3100 (2019).

Based on the above argument I do not recommend publishing the paper in *Nature Communications*. The impressive performance in the study makes the manuscript suitable for publication in a more technical journal focused on optical devices.

Technical remarks:

1. The authors stress the incoherent nature of the fluorescence light. Most studies on radiation pattern shaping deal with incoherent light. I suggest removing this focus.

2. The authors claim that their study provides a new paradigm of reconfigurable optical devices. This is strongly exaggerated and lacking credibility.
3. The three-dimensional plots in Fig. 3 do not display the data in a very clear way.
4. What is the origin of the lines visible outside the grating in Fig. 3a for the case of fluorescence light?
5. What is shown in Fig. 3f is not clearly defined in the plot or caption.
6. It is not at all clear from the text if the device can be reversibly reconfigured by just changing the humidity. From the supporting material one gets the impression that this is indeed the case.

Reviewer #2 (Remarks to the Author):

Incoherent emission steering/tunability is a very attractive topic, that has recently attracted interest in the metasurface community. Although the quantum dots-hydrogel integrated metagratings for tunable emission is interesting, we think it does not bear the right amount of novelty to warrant publication on Nature Communications. Our assessment is based on the following considerations:

- The design principle based on grating effects for control of emission directionality is not particularly novel. As the authors themselves point out in the introduction, unidirectional emission has been achieved in light-emitting metasurfaces via localized resonances, grating effect, bound states in the continuum, etc.

- The fabrication method of the metagratings was introduced by the authors in one of their previous works, *Adv. Funct. Mater.* 33, 2212053 (2023). The improvement in the current works is therefore incremental.

- Incoherent emission steering/tunability in metagratings has already been demonstrated in previous works like, "Nature Photonics 17, 588 (2023)" where the authors used an SLM, in "Nano Lett. 23, 4431 (2023)" where the authors swept the excitation position by tuning the recombination zone via external voltage and also in "Nature, 605, 447 (2022)" where the emission angle of the polariton is controlled by changing the pumping position. While the current work improves both in term of steering angle (50° vs 35°) and divergence (1.2° vs $\sim 3^\circ$), this seems an incremental improvement. Moreover, it suffers from drawbacks like speed of tuning because of the underlying mechanism of humidity control, which is inherently slow.

- It is not clear the rationale behind the use of humidity to obtain on/off operation of the device when the same functionality could be achieved by turning on and off the pump, with faster modulation rates. For the same reason, we are somewhat skeptical that the proposed scheme could have an impact in terms of applications in imaging and optical communications as mentioned by the authors.

Besides the general considerations on novelty above, we also have some technical comments, which we recommend the authors to address:

1. It would be useful if the authors could give clearer instructions on the design principles, e.g., what determines or limit the diverging angle of emission in the light-emitting metasurface?

2. Why do the authors adopt blazed gratings instead of standard gratings? The authors seem to suggest that this is for unidirectional outcoupling. Could the guided light be steered to free space via only one single diffraction order under symmetric excitation with an optimized design? or illumination condition? For example, Figure 5 seems to suggest that an important role of illumination for directional outcoupling.

3. In Fig. 2c, which diffraction order does the reflection peak correspond to? Can the authors explain why there is almost no reflection near the wavelength of PL peak? Does it mean the interaction with the grating is weak at that wavelength? Can the design be optimized?

4. It is difficult to compare angle-resolved PL (Fig. 3b and c) to angle-resolved reflection (Fig. 2f). We suggest the authors to plot them in the same way.

5. Can the authors comment why the reflection from $m=+1$ is higher than $m=+2$ while the PL emission is mainly from $m=+2$ under middle RH in Fig. 4?

Reviewer #3 (Remarks to the Author):

I co-reviewed this manuscript with one of the reviewers who provided the listed reports as part of the Nature Communications initiative to facilitate training in peer review and appropriate recognition for co-reviewers.

Reviewer #4 (Remarks to the Author):

The authors claim to demonstrate tunable unidirectional light emission from metagratings with integrated quantum dots embedded in a hydrogel matrix. Unidirectional emission is achieved by leveraging asymmetric excitation relative to the grating structure. Tunability is achieved by modifying the grating structure by swelling the hydrogel with water using humid air. The most significant results of this paper are 1) demonstration of narrow angular spread unidirectional emission using asymmetric excitation near a grating and 2) demonstrating that the unidirectional emission lobes can be switched on or off via hydrogel swelling. I recommend accepting with major revisions: the results are sufficient to warrant publication in Nature Communications provided the authors clarify some confusing and/or misleading aspects of the paper.

Major Issues:

1. I would not say that the unidirectional emission is "tunable". To me "tunable" implies a factor which is **continuously** tunable. Initially, I assumed from the title that the direction of the emission would be changing. If the authors showed a continuous variation in the emission intensity at the unidirectional emission lobe I would also be satisfied with calling it tunable. However, what they really show is that the emission lobe can be switched on or off via humidity. I recommend changing title and relevant portions of text to "switchable unidirectional emission..."

2. The authors make repeated mention of k_{parallel} (see e.g. lines 88-96, Fig. 1, Fig. 5, etc.) without ever specifying an actual **value** of k_{parallel} . Firstly, the thin film emission will not take place at a single k_{parallel} , although it likely has one or more peaks at a k_{parallel} beyond the critical momentum. Is there a single peak or multiple peak? Are these attributed to truly waveguided modes, leaky modes, or just peaks in the LDOS. How do the authors know that the unidirectional emission lobes are the 2nd order diffraction lobe? Can they give us the quantitative results that show that unidirectional lobe maps onto 2nd order diffraction from some starting feature in the unpatterned sample? Are there no k_{parallel} lobes beyond the critical angle for p-polarization? Right now this portion of the manuscript makes qualitative hand-wavy sense but lacks rigor.

3. What are the bottom and top planes of Figures 3b and 3c represent? This is not explained at all as far as I can tell. Why is this not just presented as a simple 2D color plot as in Figs. 4d, 4e, 5b, 5c, 5f, and 5g?

4. I was confused by the description of the main unidirectional peak being the 2nd order diffraction peak. My understanding is that the 1st order diffraction peak is presumed to remain beyond the critical angle, and it takes a 2nd order diffraction to bring the "guided" mode (k_{parallel} from point 2) into the escape cone. Is that correct? As an example of my confusion that I think others will also have: I think people will look at the lobe at -50 degrees in Fig. 4f and think that's 2nd order diffraction and that the peak at -25 degrees is 1st order diffraction (similar to reflection data in Fig. 4b). My understanding is that the peak in 4f at -50 degrees is 2nd order and that the peak at -25 degrees is **3rd order**. If this is correct, labeling the peaks in Figs. 4f, 5d, and 5h would be very helpful. Also, explaining things in the text more clearly, and preferably quantitatively (see point #2).

5. I think calling these "metagratings" and language such as "hydrogel film can be directly fabricated for the blazed grating profile...allowing the PL in the guided mode to be unidirectionally coupled"; "the phase gradient from the hydrogel profile imparts a grating momentum"; are misleading/confusing. In the cited works on unidirectional emission other authors use actual asymmetric phase gradient

structures. Here, as clarified later in the text and most notably in Figure 3d and Figure 5, the unidirectional emission really arises only due to asymmetric excitation on one side of the grating structure. I would argue there's not really a "phase gradient" structure per se. Certainly, they would see nearly identical behavior if they had only a "grating" that was fully symmetric. This fact is unnecessarily obscured in the presentation of the paper. It's fine to me if the authors want to call these things "metagratings", but there is really no need to. More significantly there is little justification for referring to "phase profiles", "phase gradient" or "blazed grating profile" and related language. If anything, the structure has a primarily sinusoidal "phase". I think it's more confusing than clarifying.

Minor issues:

6. Relatedly, much of the language about "momentum alteration", "momentum form hydrogel grating could be notably tuned", the "grating momentum decline" seems inaccurate. As far as I can tell from Fig. S3a, the grating momentum *does not change* upon swelling. In both cases there are 8 periods across the image. Rather, the authors are changing the grating efficiency by reducing the *amplitude* of the grating depth. This is still cool and interesting. I think the paper would be better served if this were explained more clearly.

7. It's clear there is some small asymmetry in the grating structure, but it's never really explained how/why that is chosen. If it's intentional, why didn't they fabricate a much more asymmetric blazed-grating like structure? Mostly the grating seems pretty symmetric (see e.g. Fig. 3d and 4b). If it's unintentional, why does it happen?

.

RE: Nature Communications Manuscript Revision Request (Rebuttal)

Manuscript ID: NCOMMS-23-32800

Title: Tunable unidirectional emission from quantum dots-hydrogel integrated metagratings

Author(s): Chenjie Dai, Shuai Wan, Zhe Li, Yangyang Shi, Shuang Zhang[#] and Zhongyang Li*

Reviewer #1

The manuscript reports on the optical properties of a polymer grating incorporating light-emitting quantum dots. The key claimed result is the demonstration of directional emission and the ability to change the emission pattern by changing the relative humidity. Light-emitting devices with bright directional emission in a controllable direction are key components in a wide range of applications. The topic of the present study is thus of clear importance. The use of metasurfaces, two-dimensional arrays of artificial ‘meta-atoms’, is a promising and intensively studied approach to control light emission. The topic of the study could thus have a large audience.

The study is in general well performed and the technical claims are supported by the data. The paper is mainly well written and the illustrations are clear. However, I do not agree with the claimed novelty and importance of the results.

*First of all, the authors call their structure a ‘meta-grating’ trying to place their structure in the area of metamaterials and metasurfaces. Their device is clearly a (blazed) grating and should be called such. The directional emission is based on out coupling from the polymer layer using the grating, which is a standard approach. The performance of the grating is good but there is no novel concept involved. The main novelty appears to be the use of elevated humidity to quench the grating and thus the emission. Overall, the concept is not a major and general advance of the field but rather of technical and specialized nature. Finally, there is a plethora of studies on reconfigurable gratings and metasurfaces, for example, *Nature Commun.* 9, 1396 (2018); *Small*, 2301871 (2023); *Nano Lett.* 21, 1238 (2021); *Adv. Optical Mater.* 7, 1801786 (2019); and *Science* 364, eaat3100 (2019).*

Based on the above argument I do not recommend publishing the paper in Nature Communications. The impressive performance in the study makes the manuscript suitable for publication in a more technical journal focused on optical devices.

Response: We appreciate the reviewer for carefully evaluating our manuscript and recognizing our study is “promising and intensively” and the topic with a large audience. We also sincerely thank the reviewer for providing us with such important and critical comments to help us further improve our manuscript. We have carefully improved our manuscript by following the reviewer’s comments and suggestions and hope our responses will address the questions and concerns from the reviewer.

First, and very importantly, we would try to address the main concerns on the novelty of the concept that was kindly given by the reviewer, to hopefully demonstrate why our work is high of enough novelty and impact, as well as being interesting for the broad readership of Nature Communications.

We thank the reviewer for pointing out that the usage of “metagrating” is not suitable for describing our hydrogel grating. We agree with the reviewer that the topography of our proposed hydrogel grating is a blazed grating. However, we would like to emphasize that conventional blazed grating is static while hydrogel grating exhibits extraordinary humidity-responsive optical performance transformation, so we use the term “meta-grating” to describe its active functionalities. We also recognize that the typical metagratings are composed of discrete dielectric/metallic antennas and thus may be not suitable to describe our proposed grating. To avoid any potential confusion and inappropriate expression, we have revised the expression “metagrating” to “grating” in the whole manuscript.

Regarding the reviewer’s concerns that directional emission using grating is a standard approach and thus not novel enough, we are extremely grateful that editor has allowed us the chance to highlight the novelty and innovation of this work. We agree with the reviewer that the use of gratings and photonic crystals to statically extract radiation from waveguide modes has been extensively studied due to their attractive applications. However, the active manipulation of incoherent light using reconfigurable nanophotonic devices remains a challenge, as current reconfigurable metasurfaces for coherent sources cannot be directly applied. Therefore, tunable functionalities of emission steering enabled by light-emitting nanophotonic devices are very attractive topic and are of high novelty and impact, e.g., Nat. Photonics 17, 588–593 (2023).

In this work, we originally attempted to exploit the active hydrogel materials for tunable emission steering in the far field, which is based on completely novel concepts. To the best of our knowledge, this is the first demonstration of such a hydrogel platform for far-field emission tuning. Although some gratings and metasurfaces are employed to tailor the emission direction (ACS Nano, 15, 7386-7391 (2021); Nat. Mater. 22, 1065–1070 (2023).), they provide no method of tunability due to the static materials. Whereas the use of hydrogel could serve as a new route for the realization of tunable emission steering. Since hydrogel-based nanophotonic is an emerging topic with increasingly attracted interest (Nat. Commun. 13, 6256 (2022); Sci. Adv. 8(10), eabm8598 (2022).), we believe that our proposed hydrogel grating for tunable emission is worth exploring and can gain major and valuable new learnings for the nanophotonic field to potentially expand its practical applications.

Finally, we thank the reviewer for pointing out that there are many studies on reconfigurable gratings and metasurfaces. However, the mentioned works were mostly based on coherent external sources for dynamic beam steering, rather than our proposed new route of tunable emission steering using quantum dots-integrated hydrogel grating,

which is an obvious departure. In addition, the current reconfigurable gratings and metasurfaces using laser illumination lack effective schemes for enabling emission tuning due to architecture restrictions. Therefore, we believe that our strategy using quantum dots-integrated hydrogel grating for active emission tuning is important and useful for the development of incoherent nanophotonic devices. Furthermore, we expect that this work could also act as a guideline for further work on using different active materials for tunable emission steering.

Overall, we would like to highlight that this active quantum dots-integrated hydrogel grating for tunable unidirectional emission steering is a significant and major departure from previous works, such as static grating schemes for directional emission and reconfigurable metasurfaces for coherent beam steering. Moreover, this is the very first attempt to enable such tunable emission steering with hydrogel-based nanophotonic devices, to the best of our knowledge. We apologize for not emphasizing and clarifying the novelty enough in the current manuscript format, and we have correspondingly revised the manuscript to hopefully avoid any confusion. Feel free to let us know if any other concerns.

In the manuscript:

“On the other hand, although various reconfigurable gratings and metasurfaces have been employed for dynamic beam steering with coherent external sources³⁰⁻³³, their integration with light-emitting materials has still not been fully explored.”

“Thanks to the remarkable humidity-responsive swelling capability of hydrogel materials, the grating efficiency of hydrogel grating could be notably tuned with ambient humidity change due to the morphology alteration, enabling efficient control of emission, beyond typical grating schemes for static emission extraction^{2,40}.”

Comment 1.1: *The authors stress the incoherent nature of the fluorescence light. Most studies on radiation pattern shaping deal with incoherent light. I suggest removing this focus.*

Response 1.1: We thank the reviewer for this important suggestion.

We fully agree with the reviewer that incoherence is a natural property of photoluminescence (PL). Here, we consider that Nature Communications is a multidisciplinary journal with large audiences in various fields. To accommodate readers who are not in the field of photonics, we highlight the incoherent property of emission to distinguish it from those works with coherent sources. Following the reviewer’s suggestion, we have removed the part of unnecessary emphasis on “incoherent” and revised the expression “unidirectional incoherent emission” to “unidirectional emission” in our manuscript to avoid any overclaim.

Comment 1.2: *The authors claim that their study provides a new paradigm of reconfigurable optical devices. This is strongly exaggerated and lacking credibility.*

Response 1.2: We thank the reviewer for pointing out the inappropriate overclaim.

We recognize that hydrogel materials for nanophotonic devices are an emerging strategy and have been demonstrated to enable active functionalities with coherent sources. Therefore, the claim that “provides a new paradigm of reconfigurable optical devices” might be unsuitable. However, we would like to point out that this is the first attempt to employ hydrogel nanophotonic devices for unidirectional emission tuning in this study, to the best of our knowledge. We believe that this work could act as a new guideline for future work on active light-emitting devices using different hydrogel architectures in the field of nanophotonics as well as metasurfaces, since the active control of emission using reconfigurable metasurfaces has not been fully explored. To avoid any possible overclaim, we have reorganized the related expression in the manuscript. Hope our revision could address the reviewer’s concern.

In the abstract:

“Our proposed tunable emission strategy can promote new technologies of active light-emitting devices for radiation control and optical communications.”

In the introduction:

“Such a compact and switchable unidirectional emission platform featured with simplified fabrication and easily accessible tuning strategy could facilitate new technologies of active emission tuning.”

In the conclusion:

“Our proposed strategy for tunable unidirectional emission based on hydrogel materials could serve as a promising guideline of active light-emitting metasurfaces for potential applications, including optical display, biological imaging, and incoherent light source.”

Comment 1.3: *The three-dimensional plots in Fig. 3 do not display the data in a very clear way.*

Response 1.3: We appreciate the reviewer’s constructive suggestion to help us improve the exhibition of Figure 3. We totally agree with this comment, and we have redrawn Figures 3b and 3c into two dimensions to facilitate comparison of the data.

Figure 3. Directional PL emission from Q-HIG.

Comment 1.4: What is the origin of the lines visible outside the grating in Fig. 3a for the case of fluorescence light?

Response 1.4: We thank the reviewer for bringing up the discussion on the origin of extra lines outside the grating.

According to the captured optical micrographics in Figure 3a, the visible lines come from PL intensity variation in the film and follow the grating period. This is because the grating is fabricated by the hydrogel shrinkage, and thus the film thickness is higher than the grating. When the grating is obliquely illuminated by a 532 nm laser, part of the light is coupled into the film at the end surface along the contours of the film and grating, as shown in Figure S2. Therefore, the additional coupled pump light will propagate over a distance in the film and excite fluorescence light with a grating profile. Hope our explanation would address this question. We have added the above discussion in our revised SI.

Figure S2. (a) Schematic of the fabricated grating on the hydrogel film. (b) Top view of the corresponding area of the grating under the 532-nm pump excitation.

In the supporting information:

“The captured optical micrographs in Figure 3a show that there are some visible lines outside the grating under the pump illumination. It can be observed that the visible lines come from PL intensity variation in the film and follow the grating period. This is because the grating is fabricated by the hydrogel shrinkage, and thus the film thickness is higher than the grating. When the grating is obliquely illuminated by the 532 nm laser, part of the light is coupled into the film at the end surface along the contours of the film and grating, as shown in Figure S2. Therefore, the additional coupled pump light will propagate over a distance in the film and excite fluorescence light with a grating profile.”

Comment 1.5: What is shown in Fig. 3f is not clearly defined in the plot or caption.

Response 1.5: We thank the reviewer for pointing out that the definition of order enhancement and divergence angle is missing in Figure 3f. We apologize for this mistake and have added the corresponding definition in our revised manuscript.

Here, the value of order enhancement is calculated by $I_{\text{grating}}/I_{\text{film}}$, where I_{grating} is the PL intensity of the grating at the diffractive order, and I_{film} is the PL intensity of the film at the same angle. Both intensities are measured by the same illumination condition. The divergence angle is defined by the full-width at the half-maximum of the emission angle.

In the manuscript:

“By comparing the PL intensity between the metagrating and a film at the corresponding order angle under the same illumination condition, we find an over-tenfold directional PL enhancement of Q-HIG (Figure 3f).”

“The value of order enhancement is calculated by $I_{\text{grating}}/I_{\text{film}}$, where I_{grating} is the PL intensity of the grating at the diffractive order, and I_{film} is the PL intensity of the film at the same angle. The divergence angle $\Delta\theta$ is defined by the full width at the half-maximum of the emission angle.”

Comment 1.6: *It is not at all clear from the text if the device can be reversibly reconfigured by just changing the humidity. From the supporting material one gets the impression that this is indeed the case.*

Response 1.6: We thank the reviewer for pointing out that the reconfigurable property of the proposed hydrogel grating is not highlighted enough in our manuscript. We have added the corresponding discussion about the humidity-responsive repeatability of our devices in the main text.

In the manuscript:

“In addition, we demonstrate the repeatability of the Q-HIG performance for tunable emission during cycle experiments (Figure S10). When the ambient environment around the grating cyclically alternates between middle and high RH, the PL intensity at the diffraction order could be reversibly tuned as well as other PVA-based devices^{36,43}”

Reviewer #2

Incoherent emission steering/tunability is a very attractive topic, that has recently attracted interest in the metasurface community. Although the quantum dots-hydrogel integrated metagratings for tunable emission is interesting, we think it does not bear the right amount of novelty to warrant publication on Nature Communications. Our assessment is based on the following considerations:

The design principle based on grating effects for control of emission directionality is not particularly novel. As the authors themselves point out in the introduction, unidirectional emission has been achieved in light-emitting metasurfaces via localized resonances, grating effect, bound states in the continuum, etc.

The fabrication method of the metagratings was introduced by the authors in one of their previous works, Adv. Funct. Mater. 33, 2212053 (2023). The improvement in the current works is therefore incremental.

Incoherent emission steering/tunability in metagratings has already been demonstrated in previous works like, “Nature Photonics 17, 588 (2023)” where the authors used an SLM, in “Nano Lett. 23, 4431 (2023)” where the authors swept the excitation position by tuning the recombination zone via external voltage and also in “Nature, 605, 447 (2022)” where the emission angle of the polariton is controlled by changing the pumping position. While the current work improves both in term of steering angle (50°

vs 35°) and divergence (1.2° vs ~3°), this seems an incremental improvement. Moreover, it suffers from drawbacks like speed of tuning because of the underlying mechanism of humidity control, which is inherently slow.

It is not clear the rationale behind the use of humidity to obtain on/off operation of the device when the same functionality could be achieved by turning on and off the pump, with faster modulation rates. For the same reason, we are somewhat skeptical that the proposed scheme could have an impact in terms of applications in imaging and optical communications as mentioned by the authors.

Besides the general considerations on novelty above, we also have some technical comments, which we recommend the authors to address:

Response: We appreciate the reviewer for their carefully reading and providing insightful feedback and constructive comments to help us further improve our manuscript. We thank the reviewer for recognizing our work is “interesting”. We have answered all your concerns and technical questions below, and we believe the quality and scientific content of our work has drastically improved. We hope that our revisions could address the reviewer’s concerns and convince the reviewer to support our manuscript for publication in Nature Communications.

Before responding to the reviewer’s comments on improvement and correction, we will first try to address the main concerns that were brought up, in the hope of demonstrating why our work is of high enough novelty and impact, as well as being interesting for the broad readership of Nature Communications.

First, we agree with the reviewer that the grating effects are a universal principle for emission direction control (Nat. Photonics 3, 163-169 (2009)). However, despite using grating effects, there have been various high-impact works for directional emission (Nat. Photonics 17, 588–593 (2023); Nat. Photonics 14, 543-548 (2020); Nat. Commun. 12, 3591 (2021)), and their innovation mainly comes from new methods or distinctive functions. To the best of our knowledge, our study is the very first attempt to enable tunable emission using quantum dots-integrated hydrogel grating, which has never been demonstrated before. We believe that our strategy using quantum dots-integrated hydrogel grating for active emission tuning is important and useful for the light-emitting field.

Regarding the reviewer stated their concern that the fabrication method has been demonstrated in our previous work (Adv. Funct. Mater. 33, 2212053 (2023)), we agree with the reviewer that it is not the main innovation of this work. We would like to highlight that the direct-printing technique is an effective and simple method for patterning hydrogel structure, which was originally demonstrated in our recent work. Therefore, we employ this convenient method to fabricate the proposed quantum dots-integrated hydrogel grating. Moreover, we would like to kindly point out that the

previous work focused on fabricating individual nanocavities with large pixel sizes for spectral filtering, rather than the continuous quantum dots-integrated grating for unidirectional emission in this work, which is an important difference. We expect that the direct-printing technique could serve as an effective and general fabrication method on future work for various hydrogel-based nanophotonic devices.

Regarding the relative previous works mentioned by the reviewer, we agree with the reviewer that there have been various demonstrations for emission steering/tuning, as incoherent emission manipulation is a very attractive topic. However, we intend to explore a new active strategy for unidirectional emission tuning using hydrogel materials in this work, which is a significant departure from the previous works using phase-change materials or electrical methods.

Specifically, the current work (Nat. Photonics 17, 588–593 (2023)) originally employs phase-change materials to enable tunable incoherent emission steering in sub-picosecond, and therefore was published in the high-impact journal Nature Photonics this year. However, the minimum period of phase-change grating is restricted by the SLM pixel size and leads to the limitation of maximum diffractive angle. Compared to the phase-change strategy, our proposed hydrogel grating exhibits significant improvements in unidirectional emission in terms of diffractive angle and divergence angle, as mentioned by the reviewer. We agree with the reviewer that the tuning speed of humidity control is inherently slower than electronic control due to the molecular diffusion process. Based on our previous measurement and other works (Adv. Funct. Mater. 33, 2212053 (2023); Sci. Adv. 8, eabm8598 (2022).), the hydrogel devices could respond to the humidity change within hundreds of milliseconds, which is a considerable speed in many applications. Moreover, our demonstration is more of a pioneer proof-of-concept to enable directional emission tuning using hydrogel materials. We expect that this work could offer a new approach for tunable incoherent emission steering.

In “Nano Lett. 23, 4431 (2023)”, the directional emission is induced by the angle-sensitive reflected spectra from the strong coupling of photonic BICs and excitonic resonances, rather than the emission momentum control. Moreover, the diffractive angle is less than 5° , and the divergence angle is greater than 2° . In “Nature, 605, 447 (2022)”, the authors intend to exploit BICs to enhance the lifetime of radiative excitations in a polariton waveguide, which is a departure from our work that exhibits tunable unidirectional emission. Overall, to the best of our knowledge, this is the very first attempt to enable such active unidirectional PL using quantum dots-integrated hydrogel grating with high emission performance. The above-detailed comparisons indicate a major innovative departure from the mentioned previous works. We apologize for not emphasizing and clarifying the novelty enough in the current manuscript format, and we have correspondingly revised the manuscript to hopefully avoid any confusion. Feel free to let us know if any other concerns.

Finally, the reviewer stated the concerns that the on/off function from humidity response could be equivalent to the pump source switch. We would like to kindly argue that there are many demonstrations using active methods such as liquid crystal, electrical, and liquid immersion to achieve on/off tuning, which functionalities could also be realized by switching the light source (Nat. Nanotechnol. 17, 1097-1103 (2022); Nat. Commun. 13, 1696 (2022); Light: Sci. Appl. 11, 118 (2022)). The key point is that these works provide new routes for active functionalities and could act as a guideline on future work for continuous optical tuning, and therefore have received widespread attention in the nanophotonic field as well as been published in high-impact journals. We believe that our proposed hydrogel grating with active emission tuning can promote new technologies for light-emitting metasurface. Hope our discussion would remove these concerns.

In addition, we agree with the concerns regarding the application of our hydrogel grating for incoherent emission steering in imaging and optical communications. However, as in the case with most research at the lab-scale, our work is more of a proof-of-concept idea, rather than a final product that could be directly implemented into real-world applications. Since optical imaging and communications are important application scenarios for light-emitting materials, we are confident that our tunable unidirectional emission is useful and could find potential application in the light-emitting field. For instance, according to previous work (Nat. Commun. 12, 6835 (2021)), we envision that the transmissive hydrogel grating could be integrated with a microscope for simultaneously brightfield and darkfield microscope imaging (Figure R1). Under the middle RH conditions, the PL light with a large diffractive angle could serve as the source for darkfield imaging, revealing the edge information of the sample. Meanwhile, the zero-order components of the pump light pass through the grating and provide the brightfield image of the sample. Moreover, the darkfield imaging based on PL light could be turned off by changing humidity in the cavity due to the grating effect alternation. We believe that such an active nanophotonic device for incoherent emission tuning is worth exploring and can gain major and valuable new learnings for the metasurface field to potentially expand its practical applications. Hope our discussion could address this concern.

Figure R1. Schematic of the quantum dots-hydrogel integrated grating for brightfield and tunable darkfield imaging.

Comment 2.1: *It would be useful if the authors could give clearer instructions on the design principles, e.g., what determines or limit the diverging angle of emission in the light-emitting metasurface?*

Response 2.1: We thank the reviewer's crucial suggestion to add the discussion on the design principle regarding the divergence angle of emission.

To steer the guided PL light into free space, the grating momentum k_G and in-plane momentum $k_{//}$ are required to satisfy the condition $|k_{//} + k_G|/k_0 < 1$, as mentioned in the manuscript. Therefore, the diffractive angle of out-coupled PL θ_{PL} could be expressed as $\sin\theta_{PL} = |k_{//} + k_G|/k_0 = n_{PVA}\sin\theta_i + m\lambda/P$. Since wavelength λ , period P , and refractive index n_{PVA} are constants, the divergence angle of PL $\Delta\theta_{PL}$ is major determined by the divergence angle of the incident light $\Delta\theta_{in}$. In another word, the narrow divergence angle of PL emission in this work originates from the high directional guided-mode PL in hydrogel film. In addition, the diffraction pattern could also be affected by the phase gradient deviation from fabrication and aperture diffraction from finite sample area. We have added the above discussion in our revised SI.

In the SI:

“Since wavelength λ , period P , and refractive index n_{PVA} are constants, the divergence angle of PL $\Delta\theta_{PL}$ is major determined by the divergence angle of the incident light $\Delta\theta_{in}$. In another word, the narrow divergence angle of PL emission in this work originates from the high directional guided-mode PL in hydrogel film. In addition, the diffraction pattern could also be affected by the phase gradient deviation from fabrication and aperture diffraction from finite sample area.”

Comment 2.2: *Why do the authors adopt blazed gratings instead of standard gratings? The authors seem to suggest that this is for unidirectional outcoupling. Could the guided light be steered to free space via only one single diffraction order under symmetric excitation with an optimized design? or illumination condition? For example,*

Figure 5 seems to suggest that an important role of illumination for directional outcoupling.

Response 2.2: We thank the reviewer for bringing up the question regarding the technical route of unidirectional emission.

We agree with the reviewer that the unidirectional emission is major induced by the illumination condition in this study, which could also be achieved by standard gratings. At the beginning, we intend to demonstrate asymmetric and even unidirectional PL emission under symmetric excitation using the blazed grating profile, while the diffraction of a standard grating is symmetric. Based on the design principle, if the blazed grating is perfect, the grating momentum k_G is along the single direction, and thus the guided PL light would be steered to free space with one single diffraction order under symmetric excitation. However, there are some challenges in fabricating perfect blazed gratings with current PVA direct printing technology. Therefore, we demonstrate the unidirectional emission by tailoring the direction of in-plane momentum $k_{//}$, which is determined by the illumination position. We envision that the perfect blazed hydrogel grating could be fabricated using the nanoprinting technique (Nat. Commun. 13, 6256 (2022)). In addition, we would like to highlight that the illumination position control is a convenient method to enable unidirectional emission even if the structure is a standard grating. Hope our discussion could clarify this.

Comment 2.3: *In Fig. 2c, which diffraction order does the reflection peak correspond to? Can the authors explain why there is almost no reflection near the wavelength of PL peak? Does it mean the interaction with the grating is weak at that wavelength? Can the design be optimized?*

Response 2.3: We thank the reviewer for bringing up the discussion regarding the reflection spectra. In Figure 2f, the reflection peak corresponds to the diffraction order $m = +1$ according to the grating equation $\sin\theta_r = m\lambda/P$, where θ_r is the reflection angle.

Regarding the disappearance of the reflection order near the wavelength of 575-610 nm, this phenomenon does mean that the grating ability for steering light is weak at the wavelengths. Since the optical response is highly related to the nanostructure parameter, the reflection peak wavelength will shift with the structure deviations, as shown in Figure S1. Therefore, the reflection spectra could be further optimized by designing the structure parameter and improving the fabrication precision. We have added the above discussion in our revised SI and hope our responses would address the questions.

Figure S1. (a) Measured the thickness profile of the 1800-nm period gratings with the different initial film thicknesses $t_1 = 515$ nm and $t_2 = 625$ nm. (b) Measured broadband angle-resolved reflection of the corresponding hydrogel gratings in (a) with structure parameter difference.

In the SI:

“Figure S1a shows the thickness profile of the fabricated Q-HIGs from the spin-coated film with different initial thicknesses of 515 nm (sample #1) and 625 nm (sample #2). The initial thickness difference leads to the thickness profile variation due to the different shrinkage heights. To study the impact of structural differences on optical properties, the angle-resolved reflections of hydrogel gratings are measured from sample #1 and sample #2 (Figure S1b). It can be observed that the reflection order disappears near the wavelength of 575-610 nm (sample #2), which indicates that the grating ability for steering light is weak at the wavelengths (Figure 2f). Since the optical response is highly related to the nanostructure parameter, the reflection spectra could be further optimized by designing the structure parameter and improving the fabrication precision.”

Comment 2.4: It is difficult to compare angle-resolved PL (Fig. 3b and c) to angle-resolved reflection (Fig. 2f). We suggest the authors to plot them in the same way.

Response 2.4: We thank the reviewer’s suggestion on improving the exhibition of Figures 3b and 3c. We agree with the reviewer that the 3D plots in Figure 3 is not very intuitive, and we have redrawn them into two dimensions to facilitate comparison of the data.

Figure 3. Directional PL emission from Q-HIG.

Comment 2.5: Can the authors comment why the reflection from $m = \pm 1$ is higher than $m = \pm 2$ while the PL emission is mainly from $m = \pm 2$ under middle RH in Fig. 4?

Response 2.5: We thank the reviewer for bringing up the question on the diffraction order comparison regarding reflection and PL.

Figure S4 shows the calculated diffraction order relevance between the reflection and PL emission, where the gray area indicates the wavevector component that cannot be collected by 100 \times objective. Compared to the free-space reflection under normal incident $k_{//} = 0$, the guided PL light with an in-plane wavevector $k_{//}$ induces the diffraction angle shift when coupled into the air. Specifically, the reflection order $m = \pm 1$ (dark blue/red) would be shifted out of the observable range under the PL condition due to the in-plane momentum accumulation. Therefore, although the diffraction efficiency of $m = \pm 1$ is higher than $m = \pm 2$ in reflection, only the diffraction angle of $m = \pm 2$ and ± 3 satisfy the collection condition during PL measurement. In addition, because the diffraction efficiency of $m = \pm 3$ is lower than $m = \pm 2$, the diffraction order $m = \pm 2$ is the dominant order of PL emission in the far field. To avoid any confusion, we have added the corresponding discussion in our manuscript and SI. Hope our discussion could address this question.

Figure S4. Theoretically calculated (a) reflection and (b) PL order distribution of the hydrogel grating with an 1800-nm period based on the momentum equation.

In the manuscript:

“Based on the grating momentum k_G and in-plane momentum $k_{//}$, the PL emission is identified to be the second diffraction order of the grating ($m = \pm 2$) while the first diffraction order ($m = \pm 1$) is not satisfying the outcoupling condition, and the PVA refractive index is numerically retrieved to ~ 1.52 , slightly higher than the pure PVA film's refractive index of ~ 1.51 due to QDs integration (see more details in Figures S3 and S4, Supporting Information).”

In the SI:

“Figure S4 shows the diffraction order relevance between the reflection and PL emission, where the gray area indicates the wavevector component that cannot be collected by 100 \times objective. Compared to the free-space reflection under normal incident, the guided PL light with an in-plane wavevector $k_{//}$ induces the diffraction angle shift when coupled into the air. Specifically, the reflection order $m = \pm 1$ (dark blue/red) would be shifted out of the observable range under the PL condition due to the in-plane momentum accumulation. Therefore, although the diffraction efficiency of $m = \pm 1$ is higher than $m = \pm 2$ in reflection, only the diffraction angle of $m = \pm 2$ and ± 3 satisfy the collection condition during PL measurement. In addition, because the diffraction efficiency of $m = \pm 3$ is lower than $m = \pm 2$, the diffraction order $m = \pm 2$ is the dominant order of PL emission in the far field.”

Reviewer #3

I co-reviewed this manuscript with one of the reviewers who provided the listed reports as part of the Nature Communications initiative to facilitate training in peer review and appropriate recognition for co-reviewers.

Response: We appreciate the reviewers for their effort in carefully reviewing our manuscript and providing valuable feedback and constructive comments.

Reviewer #4

The authors claim to demonstrate tunable unidirectional light emission from metagratings with integrated quantum dots embedded in a hydrogel matrix. Unidirectional emission is achieved by leveraging asymmetric excitation relative to the grating structure. Tunability is achieved by modifying the grating structure by swelling the hydrogel with water using humid air. The most significant results of this paper are 1) demonstration of narrow angular spread unidirectional emission using asymmetric excitation near a grating and 2) demonstrating that the unidirectional emission lobes can be switched on or off via hydrogel swelling. I recommend accepting with major

revisions: the results are sufficient to warrant publication in Nature Communications provided the authors clarify some confusing and/or misleading aspects of the paper.

Response: We appreciate the reviewer for carefully reading and providing valuable comments and positive feedback. We have addressed all the insightful technique questions in the following that greatly help us to improve the manuscript and clarify some important points.

Comment 4.1: *I would not say that the unidirectional emission is “tunable”. To me “tunable” implies a factor which is *continuously* tunable. Initially, I assumed from the title that the direction of the emission would be changing. If the authors showed a continuous variation in the emission intensity at the unidirectional emission lobe I would also be satisfied with calling it tunable. However, what they really show is that the emission lobe can be switched on or off via humidity. I recommend changing title and relevant portions of text to “switchable unidirectional emission...”*

Response 4.1: We thank the reviewer for pointing out the inappropriate usage regarding the expression of “tunable unidirectional emission”. We agree with the reviewer that “tunable” implies continuous control. Here, we originally intended to describe the tunable properties of the hydrogel inflation since its swelling is continuous. However, following the reviewer’s comments, we realize that the demonstrated unidirectional emission function is switched on or off via hydrogel swelling, and it does not suit the description of “tunable”. Therefore, we accept the suggestion and we have revised the corresponding expression of “tunable unidirectional emission” to “switchable unidirectional emission” in the whole manuscript, to avoid any potential confusion.

In the title:

“Switchable unidirectional emission from quantum dots-hydrogel integrated gratings”

Comment 4.2: *The authors make repeated mention of k_{parallel} (see e.g. lines 88-96, Fig. 1, Fig. 5, etc.) without ever specifying an actual *value* of k_{parallel} . Firstly, the thin film emission will not take place at a single k_{parallel} , although it likely has one or more peaks at a k_{parallel} beyond the critical momentum. Is there a single peak or multiple peak? Are these attributed to truly waveguided modes, leaky modes, or just peaks in the LDOS. How do the authors know that the unidirectional emission lobes are the 2nd order diffraction lobe? Can they give us the quantitative results that show that unidirectional lobe maps onto 2nd order diffraction from some starting feature in the unpatterned sample? Are there no k_{parallel} lobes beyond the critical angle for p-polarization? Right now this portion of the manuscript makes qualitative hand-wavy sense but lacks rigor.*

Response 4.2: We thank the reviewer for bringing up the discussion regarding the value of k_{\parallel} and the emission mode.

Firstly, according to the equation $|k_{//}| = n_{\text{PV A}} \sin \theta_i k_0 \approx k_0 n_{\text{PV A}} \sqrt{1 - (q\lambda/2n_{\text{PV A}}h)^2}$, the $|k_{//}|/k_0$ value could be calculated to 1.4334 at the wavelength of 632 nm, when the value of $n_{\text{PV A}}$ is 1.52, h is 625 nm, and the mode number q is 1. We agree with the reviewer that there are other guided modes ($q = 0, 2$) that could be coupled to the free space once beyond the critical momentum. Figure R2(a) shows the calculated angle dispersion distribution of emission at diffraction order $m = \pm 2$ for guided modes $q = 0, 1, 2$. Based on the PL measurement, it could be observed that the emission order mainly comes from the guide mode $q = 1$, while other modes are not dominant. Regarding the additional peak at the angle of $\sim 24^\circ$ in Figure 4f, it originates from the diffraction order $m = \pm 3$, as the reviewer understood in Comment 4.4. Therefore, the emission peak is major from the in-plane momentum at the mode of $q = 1$.

Regarding the origin of emission peaks, one explanation is that the intrinsic emission direction in the thin film is defined by the local density of optical states (LDOS), while the momentum of LDOS could be modified by the grating or photonic crystal effect (Nat. Photonics 14, 543–548 (2020); Nat. Photon. 17, 588–593 (2023)). However, the LDOS of emission from an unpatterned structure with an initial in-plane momentum means that the PL light is guided and propagates in the film when beyond the critical angle. Therefore, we would like to explain the phenomenon of emission peak by guided mode (Nature Photon 3, 163–169 (2009)) since the hydrogel grating is fabricated on thin film. To gain more insight into the emission, we have added the corresponding discussion in our revised SI.

Regarding the diffraction order of unidirectional emission lobes, Figure S4 shows the calculated diffraction order relationship between the reflection and PL emission, where the gray area indicates the wavevector component that cannot be collected by 100 \times objective. Compared to the free-space reflection under normal incident $k_{//} = 0$, the guided PL light with an in-plane wavevector $k_{//}$ induces the diffraction order shift when coupled into the air. Therefore, the emission angle corresponds to the diffraction order $m = 2$ of the grating effect according to the theoretical calculation. In addition, the in-plane emission from unpattern film is hard to measure based on our current setup. Since the measured emission order is well-aligned with the theoretical calculation, we think our analysis is reasonable and credible.

Regarding the p-polarization component of PL emission, Figure S9 shows the measured far-field angular dispersion of PL emission for s-polarization (S-pol) and p-polarization (P-pol). The P-pol component is more than ten times weaker than the S-pol. However, the enhanced angle-resolved PL distribution of P-pol indicates that there exists in-plane momentum for P-pol, but it is not dominant due to the 1D grating structure. We have added the corresponding discussion in our revised SI. Hope our discussion could clarify this.

Figure R2. Theoretically calculated angular emission distribution with different guided modes q and diffraction order m .

Figure S4. Theoretically calculated (a) reflection and (b) PL order distribution of the hydrogel grating with an 1800-nm period based on the momentum equation.

Figure S9. (a) Experimentally measured far-field angular dispersion of PL emission for s-polarization (S-pol) and p-polarization (P-pol) using an analyzer. The emission intensities are plotted on the same scale. (b) Measured PL intensity of P-pol in (a) with ten-fold enhancement.

In the SI:

“Figure S4 shows the diffraction order relevance between the reflection and PL emission, where the gray area indicates the wavevector component that cannot be collected by 100× objective. Compared to the free-space reflection under normal incident, the guided PL light with an in-plane wavevector $k_{||}$ induces the diffraction angle shift when coupled into the air. Specifically, the reflection order $m = \pm 1$ (dark

blue/red) would be shifted out of the observable range under the PL condition due to the in-plane momentum accumulation. Therefore, although the diffraction efficiency of $m = \pm 1$ is higher than $m = \pm 2$ in reflection, only the diffraction angle of $m = \pm 2$ and ± 3 satisfy the collection condition during PL measurement. In addition, because the diffraction efficiency of $m = \pm 3$ is lower than $m = \pm 2$, the diffraction order $m = \pm 2$ is the dominant order of PL emission in the far field.”

“Since the intrinsic emission direction in the thin film is also defined by the local density of optical states (LDOS), the momentum of LDOS could be modified by the grating or photonic crystal effect^{1,2}. However, the LDOS of emission from an unpatterned film with an initial in-plane momentum means that the PL light is guided and propagates in the film when beyond the critical angle. Therefore, we explain the phenomenon of emission peak by guided mode since the hydrogel grating is fabricated on thin film³.”

“Regarding the polarization component of PL emission, Figure S9 shows the measured far-field angular dispersion of PL emission for s-polarization (S-pol) and p-polarization (P-pol). The P-pol component is more than ten times weaker than the S-pol. However, the enhanced angle-resolved PL distribution of P-pol indicates that there exists in-plane momentum for P-pol, but it is not dominant due to the 1D grating structure.”

Comment 4.3: *What are the bottom and top planes of Figures 3b and 3c represent? This is not explained at all as far as I can tell. Why is this not just presented as a simple 2D color plot as in Figs. 4d, 4e, 5b, 5c, 5f, and 5g?*

Response 4.3: We thank the reviewer for pointing out the possible confusion regarding the exhibition of Figures 3b and 3c. Here, we plot the measured angle-resolved PL data into 2D and 3D simultaneously to show both order distribution and intensity contrast. Based on the reviewer’s comments, we realize that this presentation is quite unclear for the reader. To avoid any potential confusion, we have redrawn Figures 3b and 3c into two dimensions to facilitate comparison of the data.

Figure 3. Directional PL emission from Q-HIG.

Comment 4.4: I was confused by the description of the main unidirectional peak being the 2nd order diffraction peak. My understanding is that the 1st order diffraction peak is presumed to remain beyond the critical angle, and it takes a 2nd order diffraction to bring the “guided” mode (k_{\parallel} from point 2) into the escape cone. Is that correct? As an example of my confusion that I think others will also have: I think people will look at the lobe at -50 degrees in Fig. 4f and think that’s 2nd order diffraction and that the peak at -25 degrees is 1st order diffraction (similar to reflection data in Fig. 4b). My understanding is that the peak in 4f at -50 degrees is 2nd order and that the peak at -25 degrees is *3rd order*. If this is correct, labeling the peaks in Figs. 4f, 5d, and 5h would be very helpful. Also, explaining things in the text more clearly, and preferably quantitatively (see point #2).

Response 4.4: We thank the reviewer for pointing out the potential confusion regarding the diffraction order of the emission peak.

First, the reviewer’s understanding is completely correct. Similar to Comment 4.2, compared to the free-space reflection under normal incident $k_{\parallel} = 0$, the guided PL light with an in-plane wavevector k_{\parallel} induces the diffraction order shift (Figure S2). The 1st diffraction order would shift beyond the critical angle, while the 2nd order and 3rd order could be captured by the 100× objective. Therefore, based on the theoretical calculation, the peak in Figure 4f at -50° is 2nd order, and the peak at -25° is 3rd order. Following the reviewer’s suggestion, we have labeled the diffraction order of the peak in Figures 4f and 5d and added the corresponding discussion in our revised manuscript, to avoid any possible confusion.

Figure S4. Theoretically calculated (a) reflection and (b) PL order distribution of the hydrogel grating with an 1800-nm period based on the momentum equation.

In the manuscript:

“Based on the theoretical calculation, the emission peak at -50° is 2nd order, and the peak at -25° is 3rd order (Figure S4).”

Figure 4. Humidity control of PL radiation.

Figure 5. Unidirectional emission and radiation control.

Comment 4.5: *I think calling these “metagratings” and language such as “hydrogel film can be directly fabricated for the blazed grating profile...allowing the PL in the guided mode to be unidirectionally coupled”; “the phase gradient from the hydrogel profile imparts a grating momentum”; are misleading/confusing. In the cited works on unidirectional emission other authors use actual asymmetric phase gradient structures. Here, as clarified later in the text and most notably in Figure 3d and Figure 5, the unidirectional emission really arises only due to asymmetric excitation on one side of the grating structure. I would argue there’s not really a “phase gradient” structure per se. Certainly, they would see nearly identical behavior if they had only a “grating” that was fully symmetric. This fact is unnecessarily obscured in the presentation of the paper. It’s fine to me if the authors want to call these things “metagratings”, but there is really no need to. More significantly there is little justification for referring to “phase profiles”, “phase gradient” or “blazed grating profile” and related language. If anything, the structure has a primarily sinusoidal “phase”. I think it’s more confusing than clarifying.*

Response 4.5: We thank the reviewer for pointing out the inappropriate expression and potential confusion in the manuscript and helping us improve the quality of our manuscript.

Following the reviewer’s suggestion, we have revised the expression “metagrating” to “grating” and rewritten the corresponding sentence in the revised manuscript. Regarding the expression in terms of “phase gradient”, “phase profile”, we agree with the reviewer that the proposed hydrogel grating is more like a symmetric sinusoidal grating, and the unidirectional emission from one-side excitation could be realized by a fully symmetrical grating. We would like to kindly argue that the phase gradient is present even with sinusoidal grating topography. Moreover, based on the measured asymmetric beam steering in the far field (Figure 2f), we think our proposed hydrogel grating exists phase gradients, although not comparable to a perfect blazed grating. To avoid possible confusion, we have weakened and removed the corresponding expression regarding the blazed grating profile and related language. Please feel free let us know if any additional issues.

In the manuscript:

“By employing polymer shrinkage under the e-beam exposure, the QDs-embedded polyvinyl alcohol (PVA) hydrogel film can be directly fabricated for asymmetric grating profile with single-step processing.”

“When the in-plane emission passes through the grating, the hydrogel grating imparts a momentum $k_G = 2m\pi/P$ onto the guided modes.”

Comment 4.6: Relatedly, much of the language about “momentum alteration”, “momentum form hydrogel grating could be notably tuned”, the “grating momentum decline” seems inaccurate. As far as I can tell from Fig. S3a, the grating momentum *does not change* upon swelling. In both cases there are 8 periods across the image. Rather, the authors are changing the grating efficiency by reducing the *amplitude* of the grating depth. This is still cool and interesting. I think the paper would be better served if this were explained more clearly.

Response 4.6: We thank the reviewer for pointing out the inaccurate expression “grating momentum decline”. We agree with the reviewer that the hydrogel swelling reduces the grating efficiency rather than the momentum. We apologize for the inaccurate expression, and we have revised the related sentence to avoid any possible confusion.

Comment 4.7: It’s clear there is some small asymmetry in the grating structure, but it’s never really explained how/why that is chosen. If it’s intentional, why didn’t they fabricate a much more asymmetric blazed-grating like structure? Mostly the grating seems pretty symmetric (see e.g. Fig. 3d and 4b). If it’s unintentional, why does it happen?

Response 4.7: We thank the reviewer for bringing up the discussion on the asymmetry grating structure. We agree with the reviewer that a much more asymmetric blazed grating should be fabricated. In the beginning, we initially intended to demonstrate perfect blazed hydrogel grating for asymmetric emission and even unidirectional emission without the pump position manipulation. However, because hydrogel materials for nanofabrication have not yet been fully explored, there are some challenges in fabricating perfect blazed hydrogel gratings of this size using current PVA direct printing technology. Therefore, the fabricated grating exhibits small asymmetry. We envision that a much more asymmetric blazed hydrogel grating could be fabricated using the nanoprinting technique (Nat. Commun. 13, 6256 (2022)) or improving the material properties. We would also like to highlight that this work is the very first attempt to demonstrate the switchable emission control with asymmetric properties. To avoid any possible confusion, we have added the corresponding discussion in our revised manuscript.

In the manuscript:

“In addition, the asymmetric far-field diffraction could be further enhanced by increasing the asymmetry of the grating structure. However, because hydrogel materials for nanofabrication have not yet been fully explored, there are some challenges in fabricating perfect blazed hydrogel gratings of this size using current PVA direct printing technology. We envision that perfect blazed hydrogel gratings could be fabricated by using the nanoprinting technique or improving the material properties.”

REVIEWER COMMENTS

Reviewer #1 (Remarks to the Author):

The authors have revised the paper to address the criticisms. The changes focus on shifting the emphasis on the novel material platform and address technical remarks. After considering the changes made by the authors, I stand by my original statement that despite impressive performance the study does not present a significant novel concept, which would warrant publishing in Nature Communications. As a matter of fact, in the revised paper the claim of a tunable device has been, correctly, changed to the more modest switchable. The main novelty appears to be the novel material platform. The switching mechanism, which is based on changing the relative humidity, is not very practical. Overall I see this work better fitting in a more technical journal focused on optical materials.

Reviewer #2 (Remarks to the Author):

We thank the authors for their effort in improving the quality of the paper, adding additional information, experiments and numerical characterization. Despite the quality of the work in this submission and the thorough assessment the authors have made of the literature in their rebuttal, we remain of the idea that it does not bear the right amount of novelty to warrant publication in Nature Communications.

One of our main concerns was, and still is, that the authors make use of the key tuning mechanism (humidity infiltration in hydrogels) just to obtain an on/off operation. While the authors pointed our attention to previous works where, they say, similarly the demonstrated functionalities could be obtained by turning on/off the illumination, this is not correct. For example:

- In "Nat. Nanotechnol. 17, 1097-1103 (2022) the authors demonstrate the intensity and spectral tuning of metasurface colour pixels, spectral tunability is achieved by refractive index change and could not be achieved by turning on/off the light illumination.
- In Nat. Commun. 13, 1696 (2022) the authors multilevel optical modulation which could not be achieved by turning on/off the light illumination.
- In Light: Sci. Appl. 11, 118 (2022) the authors show color tunability of the pixels which could not be achieved by turning on/off the light illumination.

In the current manuscript, instead, the authors obtain "change of the uni-directionality of the emission" by varying the illumination position of the pump laser with respect to the grating and they obtain "on/off operation" by controlling the humidity. If they could prove directionality steering/switching by control of humidity alone, that would give an edge and justify publication in Nature Communications. The current realization, while has quite remarkable performance in terms of directionality, makes the manuscript suitable for publication in a more technical journal.

Reviewer #3 (Remarks to the Author):

I co-reviewed this manuscript with one of the reviewers who provided the listed reports as part of the Nature Communications initiative to facilitate training in peer review and appropriate recognition for co-reviewers.

Reviewer #4 (Remarks to the Author):

The authors have adequately addressed the issues raised in my prior review, I recommend publication in Nature Communications

RE: Nature Communications Manuscript Revision Request

Manuscript ID: NCOMMS-23-32800A-Z

Title: Switchable unidirectional emission from quantum dots-hydrogel integrated gratings

Author(s): Chenjie Dai, Shuai Wan, Zhe Li, Yangyang Shi, Shuang Zhang[#] and Zhongyang Li*

Reviewer #1

The authors have revised the paper to address the criticisms. The changes focus on shifting the emphasis on the novel material platform and address technical remarks. After considering the changes made by the authors, I stand by my original statement that despite impressive performance the study does not present a significant novel concept, which would warrant publishing in Nature Communications. As a matter of fact, in the revised paper the claim of a tunable device has been, correctly, changed to the more modest switchable. The main novelty appears to be the novel material platform. The switching mechanism, which is based on changing the relative humidity, is not very practical. Overall, I see this work better fitting in a more technical journal focused on optical materials.

Response: We thank the reviewer for carefully evaluating our manuscript and recognizing our study is of “impressive performance”. We are very grateful to the editor for allowing us to further enhance and clarify the novelty of our works.

Regarding the reviewer’s concerns that our novel material platform does not present a significant novel concept, we would like to kindly argue that hydrogel-based nanophotonics is an emerging and novel topic with increasingly attracted interest. Due to the remarkable deformation ability, the hydrogel materials could endow the nanophotonic devices with active tuning capability. However, the dynamic functionalities of hydrogel-based nanophotonics are currently limited to dynamic nanoprinting display or switchable holography with coherent sources (Nat. Commun. 13, 6763 (2022)).

In this work, we originally attempted to integrate hydrogel nanophotonic devices with light-emitting materials for active incoherent emission switching, which has never been demonstrated before. As the reviewer’s previous comments said “Light-emitting devices with bright directional emission in a controllable direction are key components in a wide range of applications” and “The topic of the study could thus have a large audience”, we believe that our proposed quantum dots-hydrogel integrated grating for active emission switching is worth exploring and is based on significant novel concept.

In addition, we agree with the reviewer that the switching mechanism based on humidity does not seem very practical under current conditions. However, since humidity alteration is a simple and readily available stimulus in daily life, humidity-

responsive devices have been explored for biosensing (Sci. Adv. 6(30), eabb5769 (2020)), anti-counterfeiting displays (Sci. Adv. 8, eabm8598 (2022)), and optical encryption (Nat. Commun. 13, 6763 (2022)). We are confident that our humidity-responsive scheme for switchable unidirectional emission is useful and provides a new technique for active emission switching. Moreover, as in the case with most research at the lab-scale, our work is more of a proof-of-concept idea, rather than a final product that could be directly implemented into real-world application. We envision that our device could be encapsulated into the commercialized microfluidic system (Nat. Nanotechnol. 17, 1097–1103 (2022)), which allows for precise and stable humidity control, thus enabling practical applications.

To enhance the novelty and demonstrate the potential of our hydrogel-based strategy for emission tuning, we further present a quantum dots-integrated hydrogel nanocavity for tunable directional emission. Because the emission direction cannot be tuned by humidity based on hydrogel grating schemes, we employ the angle-dependent resonance from Fabry-Perot type nanocavity to tailor the direction of PL emission (Figure 6a). With the relative humidity (RH) increase, the angle-dependent resonance actively shifts and leads to a continuous emission angle variation from $\sim 0^\circ$ to $\sim 40^\circ$ due to the cavity-induced angular enhancement of the Purcell effect and excitation rate. Figure 6b shows the measured angular-resolved PL emission under different RH conditions. The emission angle is distinctly tuned by increasing RH due to the angle-dependent absorption shift from hydrogel layer inflation (see more details in Figures S11-S14). When the RH increases from $\sim 60\%$ to 85% , the emission angle actively transforms from 0° to 40° at the emission wavelength of 575 nm (Figure 6c), revealing the tunability of the QDs-integrated hydrogel platform for continuous directional emission control. We have added the corresponding discussion in our revised manuscript in the hope of clarifying the novelty and addressing the reviewer's concerns.

Figure 6. Tunable directional emission. (a) Graphical illustration of quantum dots-hydrogel integrated nanocavity (Q-HIN) for tunable directional emission. (b) Experimentally measured far-field angular distribution of PL emission from Q-HIN under the RH of ~60%, 75%, and 85%. (c) Polar plots for corresponding angle-resolved PL intensity in (b) at the wavelength of 575 nm, as marked by dashed lines.

Overall, we believe that our proposed novel hydrogel platform for active emission switching is a completely novel concept and could have a large audience, which is also well aligned with the standards of Nature Communications.

In the manuscript:

“To prove the potential of hydrogel-based nanophotonic devices for versatile emission tuning, we further demonstrate a quantum dots-integrated hydrogel nanocavity (Q-IHN) for tunable directional emission (Figure 6a). Because the emission direction cannot be directly tuned by humidity based on hydrogel grating schemes, we employ the angle-dependent resonance from Fabry-Perot type nanocavity to tailor the direction of PL emission. With the relative humidity (RH) increase, the angle-dependent resonance actively shifts and leads to a continuous emission angle variation from $\sim 0^\circ$ to $\sim 40^\circ$ due to the cavity-induced angular enhancement of the Purcell effect and excitation rate. Figure 6b shows the measured angular-resolved PL emission under different RH conditions. The emission angle is distinctly tuned by increasing RH due to the angle-dependent absorption shift from hydrogel layer inflation. When the RH increases from $\sim 60\%$ to 85% , the emission angle actively transforms from 0° to 40° at the emission wavelength of 575 nm (Figure 6c), revealing the tunability of the QDs-integrated hydrogel platform for continuous directional emission control.”

Reviewer #2

We thank the authors for their efforts in improving the quality of the paper, adding additional information, experiments and numerical characterization. Despite the quality of the work in this submission and the thorough assessment the authors have made of the literature in their rebuttal, we remain of the idea that it does not bear the right amount of novelty to warrant publication in Nature Communications.

One of our main concerns was, and still is, that the authors make use of the key tuning mechanism (humidity infiltration in hydrogels) just to obtain an on/off operation. While the authors pointed our attention to previous works where, they say, similarly the demonstrated functionalities could be obtained by turning on/off the illumination, this is not correct. For example:

- In "Nat. Nanotechnol. 17, 1097-1103 (2022) the authors demonstrate the intensity and spectral tuning of metasurface colour pixels, spectral tunability is achieved by refractive index change and could not be achieved by turning on/off the light illumination.*
- In Nat. Commun. 13, 1696 (2022) the authors multilevel optical modulation which could not be achieved by turning on/off the light illumination.*
- In Light: Sci. Appl. 11, 118 (2022) the authors show color tunability of the pixels which could not be achieved by turning on/off the light illumination.*

Response: We thank the reviewers for their efforts in carefully reviewing our manuscript and recognizing the improvement of our manuscript. Hope our responses could address the questions and concerns from the reviewer.

The reviewer stated their main concerns that our proposed active emission tuning is an on/off operation, and therefore not novel enough. We would like to kindly argue that most previous directional emission works are static and lack tuning mechanisms. In this work, we originally attempted to integrate hydrogel nanophotonic devices with light-emitting materials for active incoherent emission tuning, which is a significant innovation compared to static schemes. Moreover, we would like to point out that the reviewers should not ignore the fact that we employ humidity to actively manipulate the grating efficiency, which could not be achieved by turning on/off the light illumination.

In addition, we agree with the reviewer that the mentioned references could achieve spectral or intensity tunability by using the refractive index, electrical, and liquid crystal schemes. However, for example, although the refractive index change is employed for spectral tuning (Nat. Nanotechnol. 17, 1097-1103 (2022)), the key operation of dynamic holography and focusing is on/off switching. Moreover, Karst et al. proposed switchable on/off beam steering using metallic polymer nanoantenna (Science 374(6567): 612-616 (2021)), which was published in the high-impact journal Science. In our opinion, the novelty of those works comes from the new route for dynamic light

control, even if the function is on/off switch. Therefore, we would like to highlight that our work originally attempted to employ the humidity-responsive deformation of hydrogel for active incoherent emission tuning, which is a completely novel concept. We envision that our switchable emission control could be further improved for continuous tuning by encapsulating the hydrogel grating into a commercialized microfluidic system with precise and continuous humidity control.

Overall, this is the very first attempt to enable such switchable unidirectional emission steering based on humidity-responsive grating efficiency alternation, to the best of our knowledge. As the reviewers previously commented “incoherent emission steering/tunability is a very attractive topic” and “the quantum dots-hydrogel integrated metagratings for tunable emission is interesting”, we believe that our proposed hydrogel grating platform for tunable emission is of sufficiently high novelty and impact in the fields of materials and nanophotonics, and conform the standards of Nature Communications.

Comment 2.1: *In the current manuscript, instead, the authors obtain "change of the uni-directionality of the emission" by varying the illumination position of the pump laser with respect to the grating and they obtain "on/off operation" by controlling the humidity. If they could prove directionality steering/switching by control of humidity alone, that would give an edge and justify publication in Nature Communications. The current realization, while has quite remarkable performance in terms of directionality, makes the manuscript suitable for publication in a more technical journal.*

Response 2.1: We thank the reviewer’s suggestion on the realization of directionality steering by only humidity control, and we appreciate the reviewer for recognizing our unidirectional emission with “quite remarkable performance”.

We agree with the reviewer that it is more interesting to achieve directionality steering and switching using humidity alone. Based on the principle of unidirectional emission, the realization of directionality emission steering requires grating phase gradient reversal. However, humidity-induced hydrogel swelling enables the grating efficiency change but not the grating gradient. Theoretically, the emission direction based on hydrogel grating schemes could only be changed by varying the pump position. Moreover, we would like to highlight that the pump position is an effective and convenient method to manipulate the wavevector of guided emission, as well as enable directionality steering. Therefore, we think that the pump position control for directionality steering tuning is an ingenious and universal approach that should not be an obstacle to publishing our work in Nature Communications.

In addition, we would like to prove the potential of hydrogel-based nanophotonic devices for continuous directional emission tuning through humidity control alone. Because the emission direction cannot be tuned by humidity based on hydrogel grating schemes, we have demonstrated a quantum dots-integrated hydrogel nanocavity for

tunable directional emission in our further study, as shown in Figure 6a. In this study, we employ the angle-dependent resonance from Fabry-Perot type nanocavity to tailor the direction of PL emission. With the relative humidity (RH) increase, the angle-dependent resonance actively shifts and leads to a continuous emission angle variation from $\sim 0^\circ$ to $\sim 40^\circ$ due to the cavity-induced angular enhancement of the Purcell effect and excitation rate. Figure 6b shows the measured angular-resolved PL emission under different RH conditions. The emission angle is distinctly tuned by increasing RH due to the angle-dependent absorption shift from hydrogel layer inflation (see more details in Figures S11-S14). When the RH increases from $\sim 60\%$ to 85% , the emission angle actively transforms from 0° to 40° at the emission wavelength of 575 nm (Figure 6c), revealing the tunability of the QDs-integrated hydrogel platform for continuous directional emission control.

To further enhance the impact and novelty of our work, we have added the above discussion regarding the tunable directional emission in our revised manuscript and supporting information.

Based on the attractive functionalities of hydrogel-based nanophotonic devices for active emission tuning, we believe that our proposed switchable unidirectional emission strategy using hydrogel nanophotonic devices could serve as a new approach for future active emission tuning works and receive widespread attention in the fields of materials and nanophotonics. We hope that our discussion could clarify the novelty and address the reviewer's concerns.

In the manuscript:

Figure 6. Tunable directional emission. (a) Graphical illustration of quantum dots-hydrogel integrated nanocavity (Q-HIN) for tunable directional emission. (b)

Experimentally measured far-field angular distribution of PL emission from Q-HIN under the RH of ~60%, 75%, and 85%. (c) Polar plots for corresponding angle-resolved PL intensity in (b) at the wavelength of 575 nm, as marked by dashed lines.

“To prove the potential of hydrogel-based nanophotonic devices for versatile emission tuning, we further demonstrate a quantum dots-integrated hydrogel nanocavity (Q-IHN) for tunable directional emission (Figure 6a). Because the emission direction cannot be directly tuned by humidity based on hydrogel grating schemes, we employ the angle-dependent resonance from Fabry-Perot type nanocavity to tailor the direction of PL emission. With the relative humidity (RH) increase, the angle-dependent resonance actively shifts and leads to a continuous emission angle variation from $\sim 0^\circ$ to $\sim 40^\circ$ due to the cavity-induced angular enhancement of the Purcell effect and excitation rate. Figure 6b shows the measured angular-resolved PL emission under different RH conditions. The emission angle is distinctly tuned by increasing RH due to the angle-dependent absorption shift from hydrogel layer inflation. When the RH increases from $\sim 60\%$ to 85% , the emission angle actively transforms from 0° to 40° at the emission wavelength of 575 nm (Figure 6c), revealing the tunability of the QDs-integrated hydrogel platform for continuous directional emission control.”

In the supporting information:

11. Humidity-responsive tunable absorption for Q-IHN

To experimentally prove the tunable angle-dependent absorption from Q-IHN, we fabricate the triple-layered Q-IHN on the silicon substrate. The QDs-integrated hydrogel layer is spin-coated on the bottom Ag mirror, and Ag layers are deposited by thermal evaporation. Due to the angle-dependent absorption, the Q-IHN's photographs exhibit distinct reflected colors under different shooting angles (Figure S11a). Figure S11b shows the measured angle-resolved reflection of Q-IHN under different humidity conditions and reveals the continuous absorption tuning due to the hydrogel layer inflation.

Figure S11. (a) Top-view photographs of the fabricated Q-IHN and bare QDs-embedded hydrogel film at normal and oblique shooting angles. (b) Measured angle-resolved reflection of the Q-IHN at different humidity conditions.

12. Measured and simulated absorption of Q-IHN at the normal incidence under different conditions.

The simulated absorption of Q-IHN with varying cavity thicknesses is plotted in Figure S12a, which is well-aligned with the measured humidity-responsive absorption alteration (Figure S12b). With the RH increasing from ~60% to ~85%, the absorption peak wavelength shift exceeds 50 nm, and the average absorption is ~70%, which confirms the excellent tunable ability from hydrogel inflation. In this design, we fabricate the top Ag layer with a thickness of ~22 nm to ensure the water molecules exchange, and the absorption could be further enhanced by optimizing the thickness and gas permeability of the top Ag layer.

Figure S12. (a) Simulated absorption of Q-IHN with different hydrogel thicknesses. (b) Corresponding measured absorption of Q-IHN under different RH.

13. Electric field distribution of Q-IHN with different hydrogel thicknesses.

To visualize the resonance variation in three-layered Q-IHN under different humidity conditions, we calculated the electric field intensity profile with the hydrogel thickness of 505 and 565 nm to mimic the hydrogel swelling from RH ~60% to RH ~85% (Figure S13). The electric field is prominently confined in the hydrogel layer when the resonance wavelength and cavity thickness satisfy the destructive interference condition, resulting in high-performance absorption.

Figure S13. Electric field distribution of Q-IHN with different hydrogel thicknesses of (i) $H = 505$ and (ii) 565 nm at the absorption peak wavelength of 568 and 628 nm.

14. Comparison of emission and reflection peaks.

Figure 14b shows the corresponding reflection shift of the Q-IHN at the wavelength of 575 nm, and the reflection dips are in excellent agreement with the angular emission peaks (Figure 14a) due to the cavity-induced resonance.

Figure S14. (a) Line plots of the corresponding angle-resolved PL intensity in Figure 6b at the wavelength of 575 nm, as marked by dashed lines. (b) Corresponding measured angle-resolved reflection of Q-IHN under different RH conditions.

Reviewer #3

I co-reviewed this manuscript with one of the reviewers who provided the listed reports as part of the Nature Communications initiative to facilitate training in peer review and appropriate recognition for co-reviewers.

Response: We appreciate the reviewers for their effort in carefully reviewing our manuscript and providing valuable feedback.

Reviewer #4

The authors have adequately addressed the issues raised in my prior review. I recommend publication in Nature Communications.

Response: We thank the reviewer for their effort in carefully reviewing our manuscript. We appreciate the reviewer for recognizing our revision and the recommendation for publication.

REVIEWERS' COMMENTS

Reviewer #1 (Remarks to the Author):

The authors have revised the manuscript and demonstrate additionally a switchable cavity. The additional data increases the impact of the work making it possible to consider it for publication in Nature Communications.

Reviewer #2 (Remarks to the Author):

We thank the authors for performing additional experiments to address some of our concerns. Through the new experiments they showed that the humidity control can be engaged somehow to change angle of emission. In order to do so, however, they had to change their grating structure.

The series of experiments conducted showed that there may be potential in the hydrogel system, however they have not reached a point of making an impactful demonstration. They showed:

- Emission intensity control by RH in hydrogel nanogratings;
- Switching angle of highly directional emission by moving the laser position in hydrogel nanogratings;
- Changing of emission directivity by RH in Ag/hydrogel/ag sandwiches (not as highly directional).

This, in our view, show that the the technology or the experiments shown so far, are not mature enough and require further improvements.

Therefore, despite the quality of the work, we remain of the idea that it does not bear the right amount of novelty to warrant publication in Nature Communications.

Reviewer #3 (Remarks to the Author):

I co-reviewed this manuscript with one of the reviewers who provided the listed reports as part of the Nature Communications initiative to facilitate training in peer review and appropriate recognition for co-reviewers.

RE: Final revisions for Nature Communications manuscript

Manuscript ID: NCOMMS-23-32800B

Title: Switchable unidirectional emission from quantum dots-hydrogel integrated gratings

Author(s): Chenjie Dai, Shuai Wan, Zhe Li, Yangyang Shi, Shuang Zhang[#] and Zhongyang Li*

Reviewer #1

The authors have revised the manuscript and demonstrate additionally a switchable cavity. The additional data increases the impact of the work making it possible to consider it for publication in Nature Communications.

Response: We sincerely thank the reviewer for the thorough evaluation of our manuscript. We appreciate the reviewer's recognition of the major revisions we made and suggestions for publication due to the increased impact of our work.

Reviewer #2

We thank the authors for performing additional experiments to address some of our concerns. Through the new experiments they showed that the humidity control can be engaged somehow to change angle of emission. In order to do so, however, they had to change their grating structure.

The series of experiments conducted showed that there may be potential in the hydrogel system, however they have not reached a point of making an impactful demonstration. They showed:

- Emission intensity control by RH in hydrogel nanogratings;
- Switching angle of highly directional emission by moving the laser position in hydrogel nanogratings;
- Changing of emission directivity by RH in Ag/hydrogel/ag sandwiches (not as highly directional).

This, in our view, show that the technology or the experiments shown so far, are not mature enough and require further improvements.

Therefore, despite the quality of the work, we remain of the idea that it does not bear the right amount of novelty to warrant publication in Nature Communications.

Response: We appreciate the reviewers for carefully examining our revised manuscript and acknowledging the improvements made, including listing our key demonstrations. We understand and respect the reviewers for their high standards and expectations when considering our work for Nature Communications.

One of the major improvements for the last-round revision is that we demonstrate to realize tunable directional emission using a quantum dots-hydrogel integrated

nanocavity, which corresponds to satisfy the reviewer's previous comment that "If they could prove directionality steering/switching by control of humidity alone, that would give an edge and justify publication in Nature Communications". We believe that our endeavors of realizing directional tunability suggested by the reviewer prove the potential of hydrogel nanophotonic devices for directionality emission steering by humidity control, thus enhancing its novelty to align with the standards for publication in Nature Communications.

We agree with the reviewer that tunable directional emission is achieved by nanocavity rather than grating. However, we would like to highlight that the proposed nanocavity strategy effectively improves the capability of hydrogel nanophotonic devices for angular emission tuning, which might turn out to be a more convenient and low-cost approach. Moreover, we would like to emphasize that our demonstrations of both grating and nanocavity are focused on hydrogel tunability with integrated quantum dots, and enable unprecedented incoherent emission tuning, thus suggesting novel platforms for studying light-matter interactions.

Regarding the reviewer's opinion that the technology might not be mature enough at the current point, we would like to kindly point out that as in the case with most research at the lab scale, our work is more of a proof-of-concept demonstration, rather than a final product that could be directly implemented into real-world application. As the reviewer's previous comments that "Light-emitting devices with bright directional emission in a controllable direction are key components in a wide range of applications", we are confident that our humidity-responsive scheme for switchable unidirectional emission is useful and promises new techniques for active emission tuning.

Overall, we have put in our best efforts and addressed all the major technical issues raised by the reviewers. We believe that the proposed hydrogel platform for active emission switching is a novel concept for attracting the attention of a large audience in the optics and nanotechnology field.

Reviewer #3

I co-reviewed this manuscript with one of the reviewers who provided the listed reports as part of the Nature Communications initiative to facilitate training in peer review and appropriate recognition for co-reviewers.

Response: We appreciate the reviewers for their efforts in carefully reviewing our manuscript and providing valuable feedback.